# TALE factors use two distinct functional modes to control an essential zebrafish gene expression program

Franck Ladam[1], William Stanney[1], Ian J Donaldson[2], Ozge Yildiz[1], Nicoletta Bobola[2], Charles G Sagerström[1]*

[1]Department of Biochemistry and Molecular Pharmacology, University of Massachusetts Medical School, Worcester, United States; [2]Faculty of Biology, Medicine and Health, University of Manchester, Manchester, United Kingdom

**Abstract** TALE factors are broadly expressed embryonically and known to function in complexes with transcription factors (TFs) like Hox proteins at gastrula/segmentation stages, but it is unclear if such generally expressed factors act by the same mechanism throughout embryogenesis. We identify a TALE-dependent gene regulatory network (GRN) required for anterior development and detect TALE occupancy associated with this GRN throughout embryogenesis. At blastula stages, we uncover a novel functional mode for TALE factors, where they occupy genomic DECA motifs with nearby NF-Y sites. We demonstrate that TALE and NF-Y form complexes and regulate chromatin state at genes of this GRN. At segmentation stages, GRN-associated TALE occupancy expands to include HEXA motifs near PBX:HOX sites. Hence, TALE factors control a key GRN, but utilize distinct DNA motifs and protein partners at different stages – a strategy that may also explain their oncogenic potential and may be employed by other broadly expressed TFs.
DOI: https://doi.org/10.7554/eLife.36144.001

*For correspondence:
Charles.Sagerstrom@umassmed.edu

## Introduction

Many transcription factors (TFs) involved in vertebrate embryogenesis are expressed across relatively large time windows that encompass a variety of cellular and morphological changes. While it seems likely that such TFs function by the same mechanism throughout embryogenesis, there is no *a priori* reason that this should be the case. One group of TFs in this category is the TALE (three amino acid loop extension) family of homeodomain proteins. The TALE family includes Pbx, as well as the closely related Prep and Meis proteins (*Waskiewicz et al., 2002*; *Deflorian et al., 2004*; *Pöpperl et al., 2000*). Pbx and Prep/Meis were originally identified as factors that form complexes with Hox TFs to drive cell fate decisions and tissue-specific gene expression starting at gastrula/segmentation stages (reviewed in [*Moens and Selleri, 2006*; *Ladam and Sagerström, 2014*; *Merabet and Mann, 2016*]). Accordingly, several Hox-dependent enhancers contain regulatory elements consisting of immediately adjacent Pbx and Hox half-sites, usually of the form TGATNNAT (*Pöpperl et al., 1995*; *Maconochie et al., 1997*; *Grieder et al., 1997*; *Ryoo and Mann, 1999*), located a short distance from TGACAG (HEXA) binding sites for Prep/Meis monomers (*Amin et al., 2015*; *Ferretti et al., 2005*; *Tümpel et al., 2007*; *Jacobs et al., 1999*; *Ferretti et al., 2000*). TALE factors also act in complexes with other tissue-specific TFs (e.g. Pdx1 (*Peers et al., 1995*), Rnx (*Rhee et al., 2004*), MyoD (*Knoepfler et al., 1999*; *Berkes et al., 2004*), Eng (*Kobayashi et al., 2003*), Otx2 (*Agoston and Schulte, 2009*) and Pax6 [*Agoston et al., 2014*]) during gastrulation/segmentation stages. Additionally, TALE factors have oncogenic potential and have been implicated in various types of leukemia (*Kamps and Baltimore, 1993*; *Nourse et al., 1990*; *Moskow et al., 1995*). In agreement with an important developmental role, disruption of TALE function leads to

severe embryonic phenotypes such that mice homozygous for null mutations in *pbx1*, *prep1* or *meis1* die *in utero,* while *pbx3* mutants die a few days after birth (*Rhee et al., 2004*; *Selleri et al., 2001*; *Fernandez-Diaz et al., 2010*; *Hisa et al., 2004*). Similarly, disruption of the earliest expressed TALE genes in zebrafish (*prep1.1*, *pbx2* and *pbx4*) produces severe embryonic defects (*Deflorian et al., 2004*; *Waskiewicz et al., 2002*; *Pöpperl et al., 2000*).

In spite of their function having been defined primarily at gastrula/segmentation stages, TALE factors are actually present throughout embryogenesis. In particular, zebrafish Prep and Pbx mRNA and protein is both maternally deposited and ubiquitously expressed in the later embryo (*Fernandez-Diaz et al., 2010*; *Deflorian et al., 2004*; *Choe et al., 2002*; *Pöpperl et al., 2000*; *Vlachakis et al., 2000*). Since all known TFs that bind TALE factors are not expressed until gastrula stages or later, it follows that TALE factors may have distinct roles prior to gastrula stages. Accordingly, Prep and Pbx can be detected at gene regulatory elements prior to the binding of their partner TFs. For instance, Prep and Pbx occupy the *hoxb1a* enhancer prior to Hoxb1b binding and before *hoxb1a* expression (*Choe et al., 2014*), while Pbx binds the *myogenin* locus before MyoD and prior to onset of *myogenin* expression (*Berkes et al., 2004*). Here, we explore the possibility that TALE factors may have uncharacterized roles during early embryogenesis. We find that maternally deposited TALE factors primarily occupy a 10 bp DECA motif at blastula stages. This motif was previously identified as a binding site for Prep:Pbx dimers (*Chang et al., 1997*; *Knoepfler and Kamps, 1997*; *De Kumar et al., 2017*; *Laurent et al., 2015*; *Penkov et al., 2013*), but was not assigned a biological role. We also find that these DECA sites have adjacent binding sites for the NF-Y pioneer TF and we show that TALE and NF-Y form a complex. Furthermore, TALE and NF-Y are required for the gradual transition to an active chromatin state of a gene network controlling anterior embryonic development. By segmentation stages, the binding repertoire of TALE factors expands to also include HEXA sites and PBX:HOX binding sites associated with the same gene network. Hence, TALE TFs control an anterior gene network throughout zebrafish embryogenesis, but do so by employing distinct DNA motifs and protein partners at different embryonic stages.

## Results

### TALE factors control a gene network regulating formation of anterior embryonic structures in zebrafish

TALE factors play a key role in early vertebrate embryogenesis, as evidenced by the phenotypes observed in TALE loss-of-function animals. In particular, loss of *prep1.1*, *pbx2* and/or *pbx4* function in zebrafish produces smaller heads and reduced eye size, as well as CNS defects – including disruptions of hindbrain segmentation – and cardiovascular defects that manifest themselves in the form of cardiac edema (*Deflorian et al., 2004*; *Waskiewicz et al., 2002*; *Pöpperl et al., 2000*), but the genetic basis of these defects is not well understood. In order to comprehensively identify TALE-dependent genes involved in embryogenesis, we used RNA-seq to compare gene expression in wildtype versus TALE loss-of-function animals. We focused on the function of *pbx2*, *pbx4* and *prep1.1* since these genes are ubiquitously expressed and represent the predominant TALE factors in the early zebrafish embryo (*Deflorian et al., 2004*; *Waskiewicz et al., 2002*; *Pöpperl et al., 2000*; *Choe et al., 2002*; *Vlachakis et al., 2000*). We used gene knock-down (KD; see *Figure 1—figure supplement 1A–C* for details) to generate embryos lacking Pbx and Prep function (as reported previously [*Waskiewicz et al., 2002*; *Deflorian et al., 2004*; *Pöpperl et al., 2000*]) and we observe the expected phenotype – including a reduced head, smaller eyes, cardiac edema, loss of pectoral fins, loss of hindbrain Mauthner neurons and disrupted cartilage formation in the head region (*Figure 1—figure supplement 1A,B*). Comparisons of RNA-seq data from control and TALE KD embryos at developmental stages (*Figure 1A*) when TALE-dependent tissues are being specified (early gastrula; 6hpf) or initiating morphogenesis (segmentation stages; 12hpf) revealed minimal gene expression changes at 6hpf (*Figure 1—figure supplement 1D–F*), but extensive changes at 12hpf (*Figure 1B*). Specifically, the expression of 671 genes (526 genes downregulated and 145 upregulated; *Figure 1C*) is altered in TALE KD embryos compared to control embryos at 12hpf. GO-term analysis on the genes downregulated in 12hpf TALE KD embryos revealed an enrichment for roles in embryonic development – particularly head formation, neural development (including eye and hindbrain development) and circulatory system formation (*Figure 1D*), consistent with the TALE

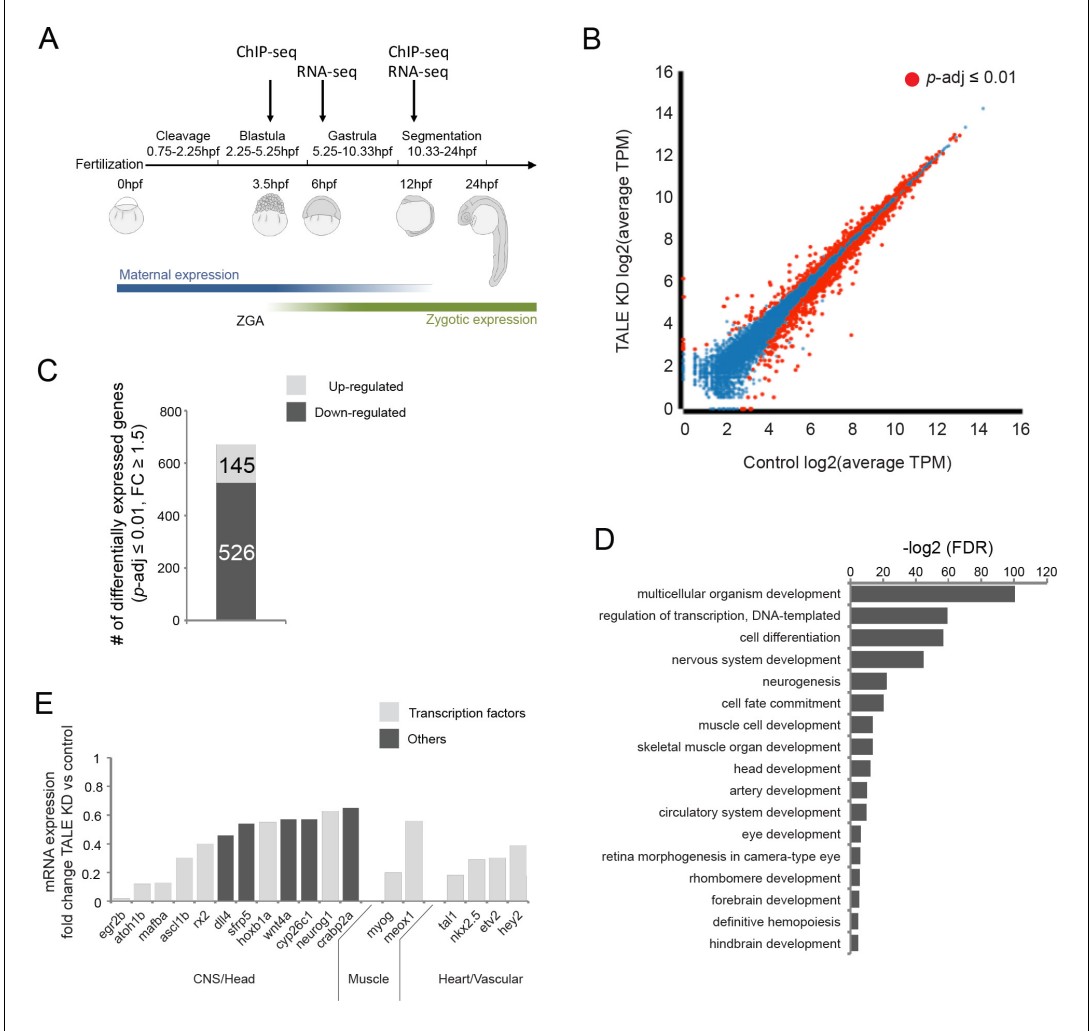

**Figure 1.** TALE factors control a gene network regulating formation of anterior embryonic structures. See also *Figure 1—figure supplement 1*. (**A**) Schematic of zebrafish embryogenesis indicating time points used for RNA-seq and ChIP-seq analyses. The 3.5hpf time point represents a stage prior to robust zygotic gene expression, while 12hpf corresponds to the time when tissue morphogenesis is initiated. The 6hpf time point for RNA-seq was selected to capture changes in gene expression occuring shortly after ZGA. ZGA = zygotic genome activation; hpf = hours post-fertilization. (**B**) Scatter plot showing average TPM gene expression as identified by RNA-seq in control vs TALE KD 12hpf embryos. Genes with significant expression variation (p-adj ≤0.01) are highlighted in red. Statistical test = Wald test in DeSeq2. (**C**) Graph showing the number of genes up/downregulated (p-adj ≤0.01, fold-change ≥1.5) in 12hpf TALE KD samples vs control. (**D**) DAVID analysis of genes downregulated (p-adj ≤0.01, fold-change ≥1.5) in 12hpf TALE KD samples vs control. Note that only select categories are presented, a full list of GO terms is available in *Supplementary file 3*. FDR = Benjamini multiple testing False Discovery Rate. (**E**) Expression fold-change of select genes significantly downregulated in 12hpf TALE KD samples compared to control. Genes were selected based on their role in regulation of relevant embryonic structures.

DOI: https://doi.org/10.7554/eLife.36144.002

The following figure supplement is available for figure 1:

**Figure supplement 1.** Characterization of the TALE KD phenotype.

DOI: https://doi.org/10.7554/eLife.36144.003

KD phenotype. Furthermore, these TALE-regulated genes are enriched for transcriptional regulators and a large number encode known TFs (*Figure 1D,E*), suggesting that this gene set defines a gene regulatory network (GRN). Upon comparison to previously reported TALE loss-of-function phenotypes, we find that of 13 Pbx-dependent genes identified in the zebrafish retina and hindbrain (*French et al., 2007*), seven (*egr2b, mafba, eng2b, rx2, gdf6a, hmx4, meis3*) are also downregulated in our analysis. Similarly, of six genes downregulated in Prep loss-of-function zebrafish (*Deflorian et al., 2004*), four (*pax6a, hoxb1a, hoxa2b, hoxb2a*) are downregulated in our

experiment. This suggests that our RNA-seq analysis captured a comprehensive set of TALE-dependent genes. We conclude that TALE TFs control a gene regulatory network (TALE GRN), which instructs anterior embryonic development and that becomes operative between 6hpf and 12hpf.

## Genomic TALE occupancy is continuously and dynamically associated with the TALE GRN during embryogenesis

To determine how genomic TALE occupancy relates to the TALE GRN, we carried out ChIP-seq for Prep1.1 in zebrafish embryos. We assessed TALE binding both at 12hpf (early segmentation stage; when TALE-dependent gene expression is detectable; *Figure 1A,B*), and also at 3.5hpf (late blastula stage; prior to robust zygotic gene expression; *Figure 1A*, *Figure 2—figure supplement 1A,B*). Analysis of two biological replicates at each stage (using a cutoff of FE $\geq$ 10; *Figure 2A*, *Supplementary file 1*) yielded ~13,300 peaks at 3.5hpf ($Prep_{3.5hpf}$) and ~24,200 peaks at 12hpf ($Prep_{12hpf}$), the majority of which are located within 30 kb of a transcription start site (TSS; *Figure 2B*). We note that out of the 13,300 $Prep_{3.5hpf}$ peaks, ~60% co-localize with a $Prep_{12hpf}$ peak (*Figure 2C*), suggesting that a large fraction of binding sites remains occupied throughout embryogenesis. However, an additional ~16,500 peaks detectable at 12hpf do not co-localize with a $Prep_{3.5hpf}$ peak, demonstrating that additional binding sites become occupied at later stages. We refer to binding sites observed only at 12hpf as '12hpf-only' ($Prep_{12hpf-only}$). We noticed that although the $Prep_{12hpf-only}$ peaks do not co-localize with $Prep_{3.5hpf}$ peaks, the two types of sites nevertheless appear to be preferentially located near one another (*Figure 2A*). Indeed, a quantitative analysis of peak distribution revealed that 58% of all $Prep_{12hpf-only}$ peaks are located within 40 kb of a $Prep_{3.5hpf}$ peak (*Figure 2D*, *Figure 2—figure supplement 1C*).

GO-term analyses revealed that genes associated with either $Prep_{3.5hpf}$ or $Prep_{12hpf-only}$ peaks are enriched for functions related to transcriptional regulation and embryonic development – particularly neural development, but also heart and muscle formation (*Figure 2E*). These functions correspond well with the phenotype observed in TALE KD embryos (*Figure 1—figure supplement 1A,B*) and with the GO-terms associated with the TALE GRN (*Figure 1D*), suggesting that Prep occupancy is linked with the TALE GRN throughout embryogenesis. Accordingly, we find that ~70% (350/526) of the TALE GRN genes are located within 30 kb of a $Prep_{3.5hpf}$ or a $Prep_{12hpf-only}$ peak (*Figure 2F*).

We conclude that Prep occupies genomic binding sites associated with the TALE GRN as early as late blastula stages. ~60% of these sites are also occupied at segmentation stages, but by this stage a large number of additional binding sites ($Prep_{12hpf-only}$ sites) have become bound by Prep. Since these later sites are also associated with the TALE-GRN, Prep binding is dynamically and continuously associated with the TALE GRN during zebrafish embryogenesis.

## TALE factors utilize distinct binding motifs at early versus late stages of embryogenesis

The widespread genomic binding of Prep at blastula stages has not been reported previously and we therefore examined the characteristics of these binding sites in greater detail. To this end, we used the MEME de novo motif discovery tool (*Bailey et al., 2009*; *Machanick and Bailey, 2011*) and identified a 10 bp TGATTGACAG sequence as the predominant motif centered at $Prep_{3.5hpf}$ peak summits (*Figure 3A*). This 'DECA motif' contains immediately adjacent Pbx and Prep half sites and was initially identified as a binding site for TALE dimers in vitro (*Chang et al., 1997*; *Knoepfler and Kamps, 1997*). Subsequently, the DECA motif has been detected at sites co-occupied by Pbx and Prep in embryonic stem cells and in the mouse trunk (*Laurent et al., 2015*; *Penkov et al., 2013*; *De Kumar et al., 2017*), but it has not been assigned a biological function. To test if DECA sites are co-occupied by Pbx also in the zebrafish embryo, we selected twelve binding sites and used ChIP-qPCR to assay Pbx occupancy. We find that Pbx is present at eleven of the twelve sites at 3.5hpf and that all twelve are occupied by Pbx at 12hpf (*Figure 3C*), revealing that Prep and Pbx co-occupy DECA sites at least through segmentation stages.

Notably, the DECA motif detected at $Prep_{3.5hpf}$ peaks is distinct from the typical configuration of binding motifs recognized by TALE factors in their role as cooperating with tissue-specific TFs (reviewed in [*Ladam and Sagerström, 2014*; *Merabet and Mann, 2016*]). Since this role was characterized primarily at segmentation stages (*Ferretti et al., 2005*, *2000*; *Jacobs et al., 1999*; *Tümpel et al., 2007*; *Pöpperl et al., 1995*), we considered the possibility that the $Prep_{12hpf-only}$

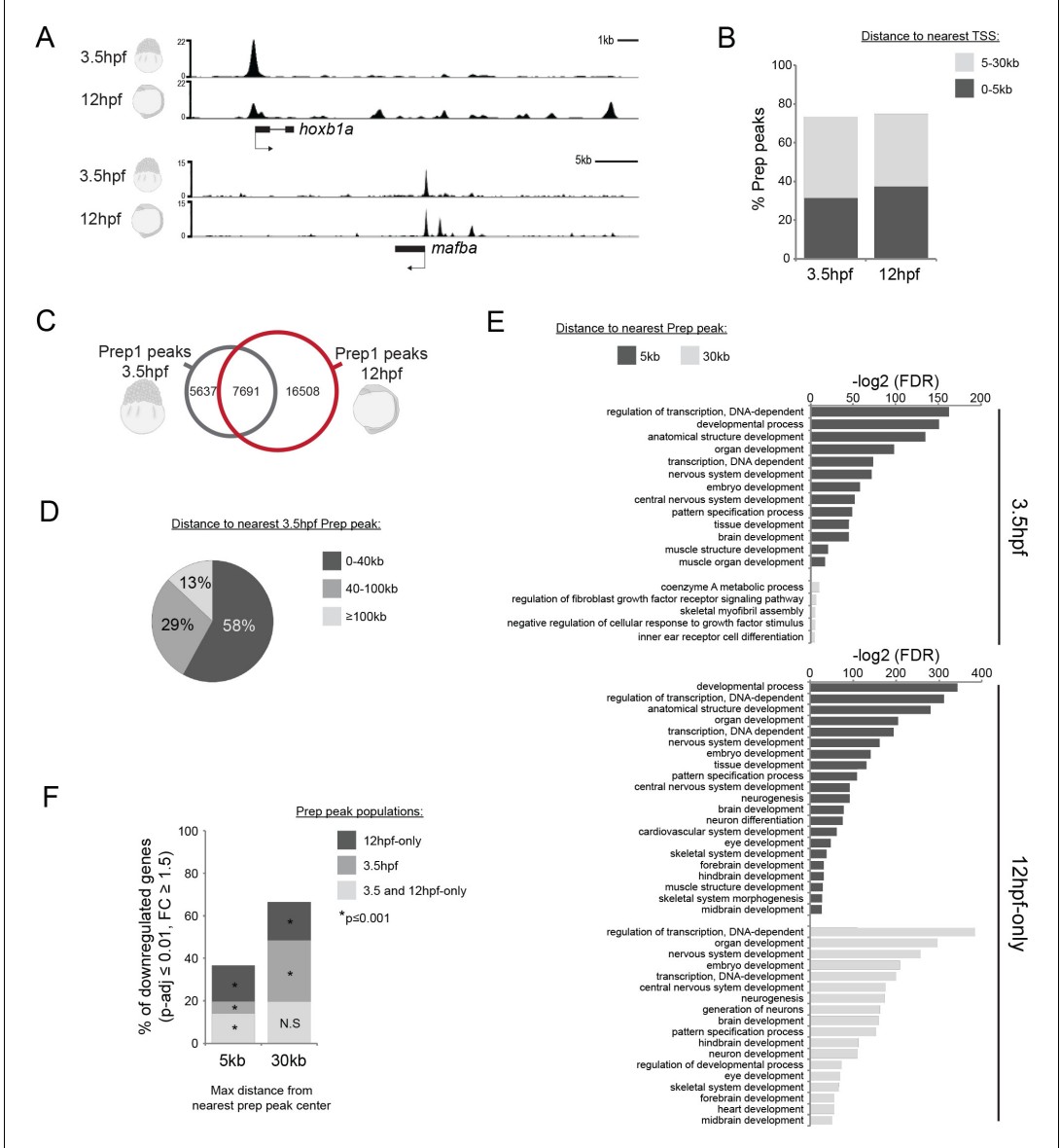

**Figure 2.** Genomic TALE occupancy is continuously and dynamically associated with the TALE GRN during embryogenesis. See also *Figure 2—figure supplement 1*. (**A**) Representative UCSC browser tracks illustrating Prep binding at the *hoxb1a* and *mafba* loci in 3.5 and 12hpf embryos. (**B**) Graph showing the distribution of $Prep_{3.5hpf}$ and $Prep_{12hpf}$ binding sites relative to TSSs. (**C**) Venn diagram illustrating co-localization of Prep peaks in 3.5hpf and 12hpf embryos. Two peaks are considered to co-localize if their summits are within 50 bp. (**D**) Chart illustrating percent of $Prep_{12hpf-only}$ peaks found at various distances from $Prep_{3.5hpf}$ peaks. (**E**) GO term enrichment for $Prep_{3.5hpf}$ and $Prep_{12hpf-only}$ peaks identified by GREAT using the nearest gene within 5 or 30 kb association rule. In the case of GO terms associated with genes within 30 kb, only select categories are presented, a full list of GO terms is available in *Supplementary file 3*. FDR = Binomial False Discovery Rate. (**F**) Graph showing percent of TALE GRN genes (p-adj ≤0.01, fold-change ≥1.5) associated (≤5 or 30 kb) with $Prep_{3.5hpf}$ and $Prep_{12hpf-only}$ peaks. *p*-values for enrichment above a random set of genes were calculated using the Pearson correlation test.

DOI: https://doi.org/10.7554/eLife.36144.004

The following figure supplement is available for figure 2:

**Figure supplement 1.** Analysis of TALE binding in zebrafish embryos.
DOI: https://doi.org/10.7554/eLife.36144.005

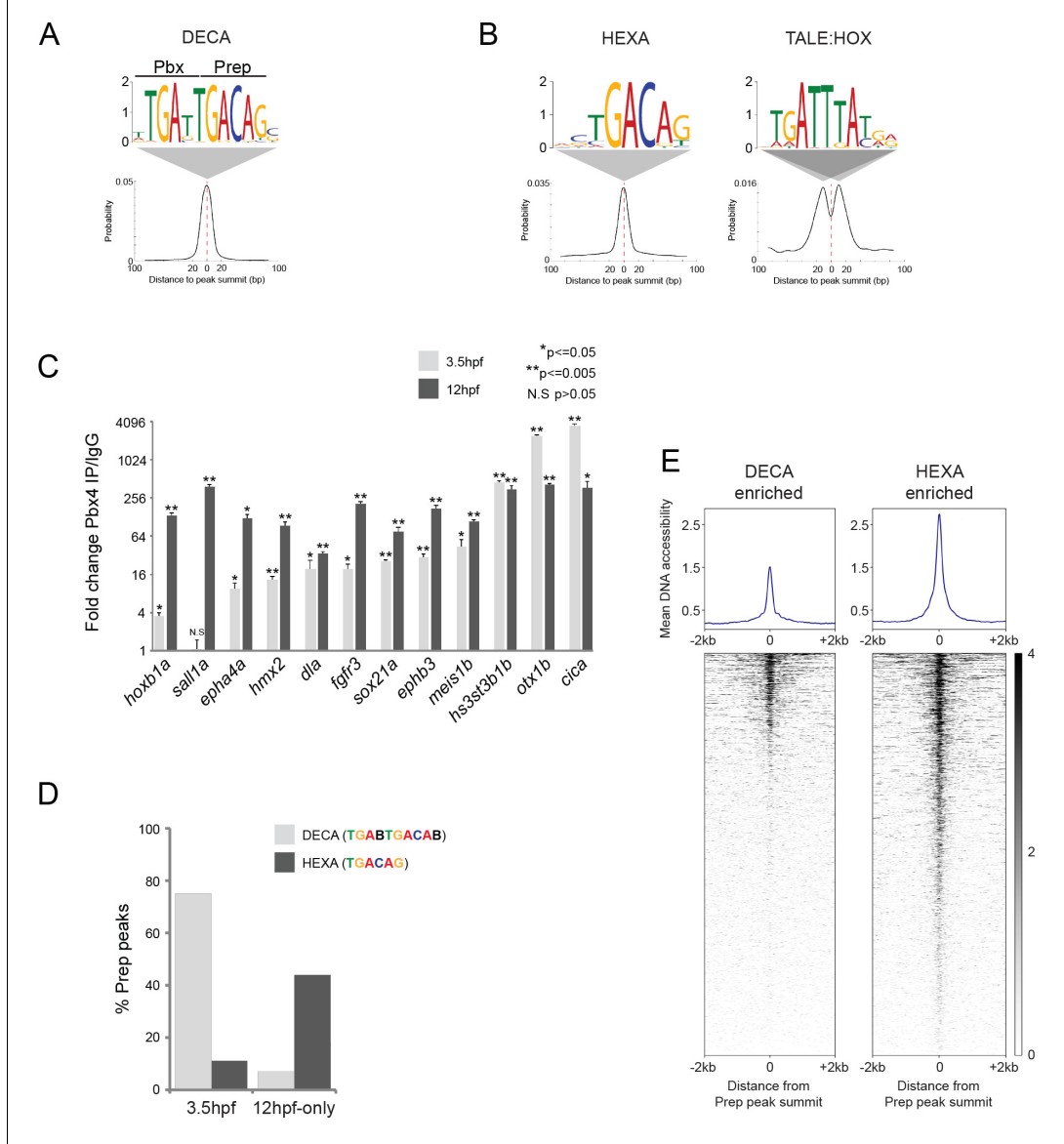

**Figure 3.** TALE factors utilize distinct binding motifs at early versus late stages of embryogenesis. (**A**) Sequence logo and localization relative to Prep peak summits of sequence motifs identified by MEME at Prep$_{3.5hpf}$ peaks. (**B**) Sequence logo and localization relative to Prep peak summits of sequence motifs identified by MEME at Prep$_{12hpf-only}$ peaks. (**C**) ChIP-qPCR showing Pbx4 binding at Prep-occupied DECA sites at 3.5hpf and 12hpf, labeled with the name of the nearest gene. Data of three independent biological replicates are presented as mean fold change ± SEM of Pbx4 IP vs control IgG. Statistical test: unpaired t-test. (**D**) Graph showing percent of Prep$_{3.5hpf}$ and Prep$_{12hpf-only}$ peaks that contain DECA or HEXA motifs. (**E**) Heatmaps displaying chromatin accessibility at 4hpf (derived from ATAC-seq data [*Kaaij et al., 2016*]) at DECA (left panel) and HEXA (right panel) enriched peaks. (Prep$_{3.5hpf}$ and Prep$_{12hpf-only}$ peaks were used as a source of DECA- and HEXA-enriched sites, respectively.).

DOI: https://doi.org/10.7554/eLife.36144.006

The following source data is available for figure 3:

**Source data 1.** Input sequences.
DOI: https://doi.org/10.7554/eLife.36144.007

peaks may represent TALE factors acting together with tissue-specific TFs. Indeed, MEME analysis of Prep$_{12hpf-only}$ peaks returned a 6 bp TGACAG (HEXA) motif, but not the DECA motif (*Figure 3B*). HEXA motifs are binding sites for monomeric Prep (or Meis) factors (*Chang et al., 1997*; *Berthelsen et al., 1998*; *Shen et al., 1997a*) and have been found at several Hox-dependent regulatory elements (*Amin et al., 2015*; *Ferretti et al., 2000*; *Ryoo et al., 1999*; *Jacobs et al., 1999*;

*Tümpel et al., 2007*). Accordingly, MEME also identified a TGATTTAT sequence, which represents a binding site for TALE:HOX dimers (*Penkov et al., 2013*; *Shen et al., 1997b*; *Chang et al., 1996*), at the Prep$_{12hpf-only}$ peaks (*Figure 3B*). This Hox motif is not located at the center of the Prep peaks, but is off-set by ~10 bp, as has been observed previously at regulatory elements where Prep/Meis acts with Hox TFs (*Jacobs et al., 1999*; *Ferretti et al., 2005*, *2000*). We next examined the prevalence of the different motifs at Prep$_{3.5hpf}$ versus Prep$_{12hpf-only}$ peaks. We find that 75% of Prep$_{3.5hpf}$ binding sites contain a DECA motif, while only 7% of Prep$_{12hpf-only}$ sites do so. Conversely, 44% of all Prep$_{12hpf-only}$ binding sites, but only 11% of Prep$_{3.5hpf}$ sites, contain a HEXA motif (*Figure 3D*). Consistent with HEXA motifs being associated with a Prep cofactor role, we also find that PBX:HOX binding sites are more prevalent at Prep$_{12hpf}$ peaks (24%) than at Prep$_{3.5hpf}$ peaks (5%). It is surprising that HEXA sites are not occupied by Prep at blastula stages and we considered the possibility that HEXA sites may not be accessible at this stage. We made use of previously published ATAC-seq data (*Kaaij et al., 2016*) to examine DNA accessibility at DECA versus HEXA sites at 4hpf and find that HEXA sites are considerably more accessible than DECA sites (*Figure 3E*), suggesting that chromatin accessibility is not a limiting factor for Prep binding at HEXA sites in the blastula stage embryo.

While both DECA and HEXA sites have been reported previously, our data show for the first time that there is a temporal order to how TALE factors utilize these motifs during embryogenesis. Specifically, TALE factors occupy primarily DECA sites at blastula stages and these motifs remain occupied at least until segmentation stages, but by segmentation stages additional binding sites become utilized so that TALE factors also occupy HEXA motifs associated with binding sites for tissue-specific TFs such as Hox proteins.

## Some TALE-occupied sites are associated with chromatin marks at Blastula stages

Previous analyses of individual DNA elements containing HEXA motifs adjacent to PBX:HOX motifs demonstrated that these act as enhancers in mouse and zebrafish (*Pöpperl et al., 1995*; *Jacobs et al., 1999*; *Ferretti et al., 2005*; *Choe et al., 2009*; *Ferretti et al., 2000*; *Di Rocco et al., 1997*; *Manzanares et al., 2001*; *Tümpel et al., 2007*; *Wassef et al., 2008*). Conversely, de novo motif discovery in conserved hindbrain enhancers – combined with functional testing in zebrafish – identified HEXA and PBX:HOX motifs as being essential for enhancer activity (*Parker et al., 2011*; *Grice et al., 2015*). Accordingly, we find that the Prep$_{12hpf-only}$ peaks are found at highly conserved regions of the genome (*Figure 4—figure supplement 1A*) and are associated with chromatin modifications known to mark enhancers (*Figure 4—figure supplement 1B*). Finally, we find that of 74 hindbrain enhancers active at 48–72hpf (*Grice et al., 2015*), 19 (26%; *Figure 4—figure supplement 1C*) are associated with a Prep$_{12hpf-only}$ peak. Hence, the arrangement of HEXA sites associated with PBX:HOX motifs (and other tissue-specific TF motifs) that we observe at 12hpf is very likely to represent enhancer elements.

In contrast, no biological function has yet been assigned to elements containing DECA motifs. We characterized 11 Prep-occupied DECA sites in greater detail and find that eight are associated with genomic regions conserved in five other fish species (*Figure 4—figure supplement 1D*). Six of these elements are also conserved in mammals, suggesting that they play an evolutionarily important role. To identify a role for these elements, we tested whether Prep$_{3.5hpf}$ peaks correlate with particular chromatin features by comparison to available ChIP-seq data sets from 4.5hpf blastula stage zebrafish embryos (*Bogdanovic et al., 2012*; *Zhang et al., 2014*; *Lee et al., 2015*). Ranking TALE-bound regions based on their level of H3K4me1 (a histone modification associated with enhancers and promoters) reveals a clear pattern (*Figure 4A*). In particular, K-means clustering produced four clusters of sequences, three of which (representing ~25% of all TALE-occupied sites) are highly marked by H3K4me1. To distinguish TALE-occupied sites associated with chromatin marks from sites that lack (or display very low levels of) such marks, we refer to them as MPADs (Modified Prep Associated Domains) and non-MPADs, respectively. We find that MPADs are also enriched for H3K4me3 (a mark of active promoters) and H3K27ac (a mark of active enhancers and promoters). In addition, MPADs center on nucleosome-depleted regions and are highly enriched for RNA polymerase II occupancy (*Figure 4A,B*). MPADs are also preferentially found within 5 kb of TSSs (*Figure 4C*), are enriched near genes involved in transcriptional regulation and embryonic development (*Figure 4D*, *Supplementary file 2*) and are found at conserved sites in the genome (*Figure 4E*). In contrast, the

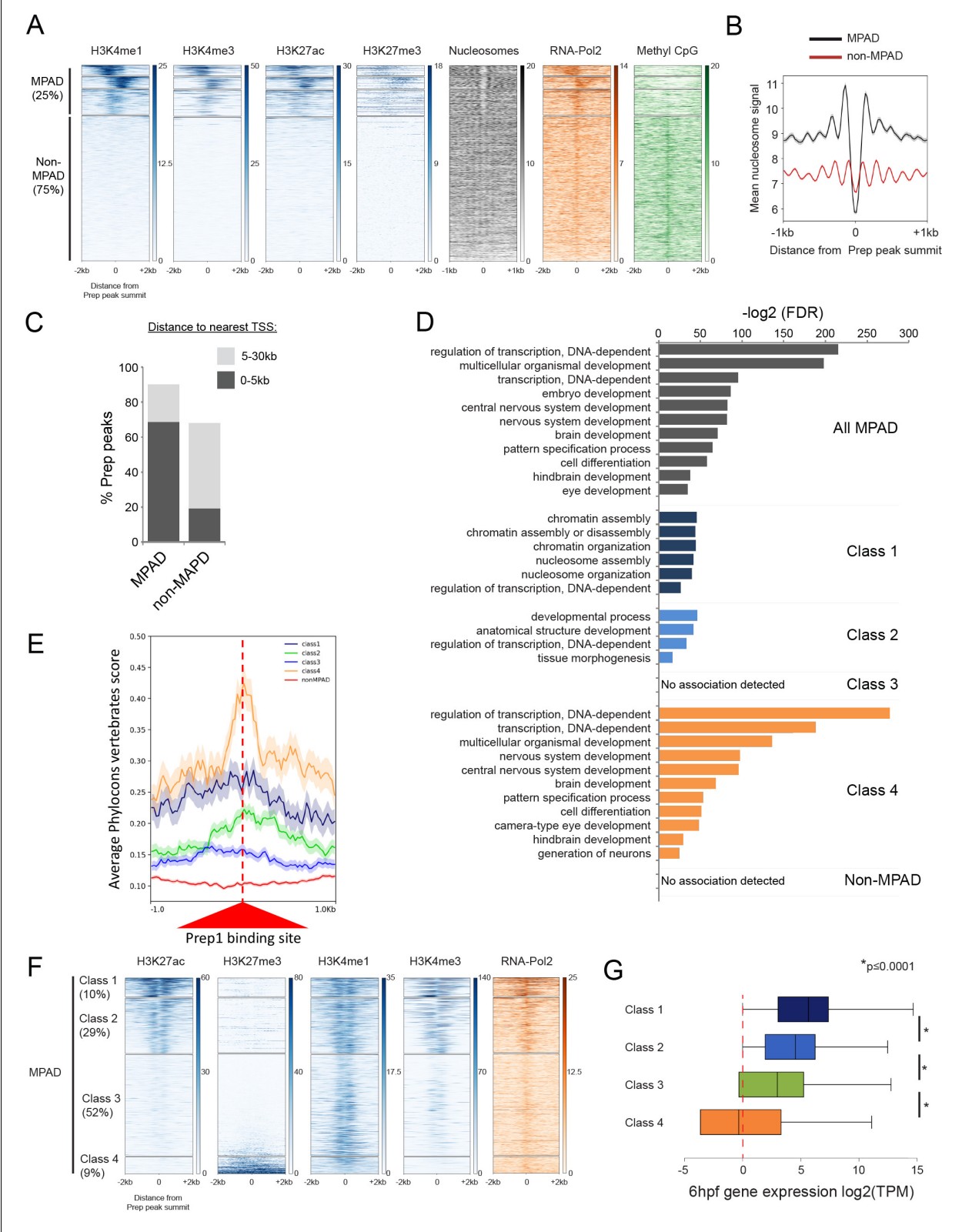

**Figure 4.** Some TALE-occupied sites are associated with chromatin marks at blastula stages and developmental control genes are enriched near MPADs displaying repressive histone modifications. See also *Figure 4—figure supplements 1* and *2*. (**A**) Heatmaps displaying chromatin features at genomic regions occupied by Prep at 3.5hpf. H3K4me1 signals at Prep-occupied elements was analyzed by K-mean (k = 4) clustering (left panel). H3K4me3, H3K27ac, H3K27me3, nucleosome, RNA-pol2 subunit RPB1 and Methyl CpG signals are displayed based on the H3K4me1 clustering order.
*Figure 4 continued on next page*

*Figure 4 continued*

(B) Average nucleosome signal at MPADs and non-MPADs (as defined in A). (C) Distribution of MPADs and non-MPADs relative to TSSs. (D) GO term enrichment for MPADs and non-MPADs identified by GREAT (nearest gene within 30 kb). Note that genes associated with Class 3 MPADs or non-MPADS are not enriched for GO terms. Only select categories are presented, a full list of GO terms is available in **Supplementary file 2**. FDR = Binomial False Discovery Rate. (E) Conservation of 3.5hpf Prep-occupied sites among vertebrates generated using PhastCons vertebrate 8-way comparison. The score shown is the probability ($0 \leq p \leq 1$) that each nucleotide belongs to a conserved genomic element. (F) Heatmaps displaying chromatin features at MPADs. H3K27ac and H3K27me3 signals at MPADs were analyzed by K-mean (k = 4) clustering. H3K4me1, H3K4me3, nucleosome and RBP1 signals are displayed based on the H3K27ac/me3 clustering order. (G) Box plots showing average expression of genes near (≤30 kb) each of the four MPAD classes, as determined by RNA-seq on 6hpf embryos. Data are presented as log2 of mean TPM (transcripts per million) values from three biological replicates. Statistical test: pairwise comparison with Kruskal-Wallis followed by Dunn's post-hoc test.
DOI: https://doi.org/10.7554/eLife.36144.008

The following figure supplements are available for figure 4:

**Figure supplement 1.** Comparison of early and late Prep-occupied sites.
DOI: https://doi.org/10.7554/eLife.36144.009

**Figure supplement 2.** TALE occupancy in mESCs is associated with chromatin profiles similar to 3.5hpf zebrafish embryos.
DOI: https://doi.org/10.7554/eLife.36144.010

remaining 75% of TALE-occupied sites display only sparsely modified histones at this stage (**Figure 4A**). These non-MPAD sites lack a nucleosome free region (**Figure 4B**) and are only weakly associated with RNA Polymerase II, but they are highly methylated on CpG dinucleotides. The non-MPAD sites are mostly found at distances greater than 5 kb from TSSs (**Figure 4C**), associated genes are not enriched for any specific functions (**Figure 4D**) and they are not highly conserved (**Figure 4E**).

Prep occupancy has not been assessed in blastula stage embryos of other animal species, but previous analyses in murine embryonic stem cells (mESCs) identified Prep as bound to DECA motifs ([**Laurent et al., 2015**]; see also **Figure 4—figure supplement 2A**). We find that ~40% (1595/4008) of the Prep-associated genes in mESCs have orthologs with a nearby $Prep_{3.5hpf}$ peak in zebrafish (**Figure 4—figure supplement 2B,C**), indicating that Prep binding near developmental control genes is evolutionarily conserved. Sorting Prep-occupied regions from mESCs based on their enrichment for H3K4me1 revealed characteristics similar to those observed in zebrafish (**Figure 4—figure supplement 2D,E**), although there are many fewer unmodified regions in mESCs than in zebrafish embryos. Hence, at blastula stages, TALE-occupied sites can be divided into ones that are associated with various chromatin marks and are located near promoter regions of developmental control genes (MPADs), and ones that are largely devoid of histone marks and that are not associated with specific gene functions (non-MPADs).

## Developmental control genes are enriched near MPADs displaying repressive histone modifications

We noticed that a subset of MPADs shows detectable enrichment for the repressive H3K27me3 histone modification (**Figure 4A**). To examine this finding further, we ranked MPADs based on their level of H3K27ac and H3K27me3 at blastula stages. K-means clustering divided the resulting distribution into four groups (**Figure 4F**). For the sake of comparison, we refer to these as Class 1–4 MPADs. In particular, MPADs with high (Class 1) and intermediate (Class 2) levels of H3K27ac are associated with high levels of H3K4me3 and RNA Pol II occupancy, while elements with low levels of H3K27ac (Class 3 and 4) are not. Notably, the subset of MPADs with the lowest level of H3K27ac are associated with high levels of H3K27me3 (Class 4). When we analyze the GO-terms of genes associated with each of the four MPAD classes, we find that H3K27me3-modified Class 4 MPADs are more highly associated with developmental control genes than are Class1-3 MPADs (**Figure 4D**). In agreement with the chromatin profile at MPADs, RNA-seq analysis at 6hpf (shortly after the onset of zygotic gene expression) revealed that genes associated with Class 1 and 2 MPADs are expressed at higher levels than genes associated with Class 3 and 4 MPADs (**Figure 4G**). Similarly, ranking MPADs from mESCs based on H3K27ac levels revealed categories analogous to those observed in zebrafish (**Figure 4—figure supplement 2F,G**).

Hence, MPADs can be further subdivided such that Class 1 and 2 display active chromatin marks and are found near genes expressed at 6hpf. In contrast, Class 4 MPADs are marked by H3K27me3

and are associated with genes involved in developmental processes, but these are not highly expressed at 6hpf. Class 3 MPADs are only marked by H3K4me1 and genes associated with this class show an intermediate level of expression at 6hpf, but they are not enriched for specific biological functions. We conclude that the chromatin state of MPADs correlates with the biological function of nearby genes and that developmental control genes are primarily associated with repressed (H3K27me3-modified) Class 4 MPADs.

## Class 4 MPADs transition to an active chromatin state during embryogenesis

We next examined whether chromatin modifications at MPADs change as embryogenesis progresses by comparing their H3K27ac status at the blastula stage (4.5hpf) to that at late gastrula (9hpf) – when the embryonic axes have formed and organogenesis is beginning. We find that Class 1 and 2

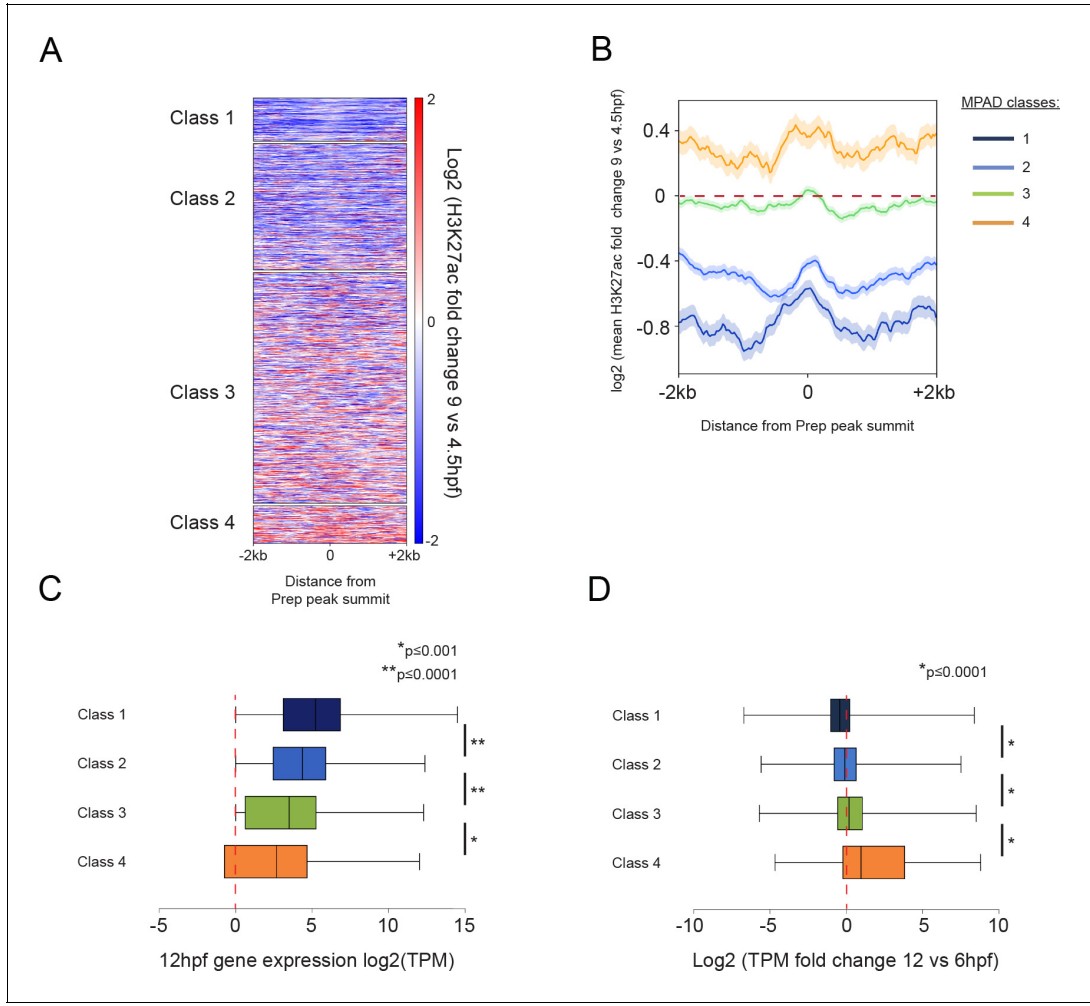

**Figure 5.** Class 4 MPADs transition to an active chromatin state during embryogenesis. See also *Figure 5—figure supplement 1*. (A) Heatmap displaying the change in H3K27ac signal (log2 of fold-change) at MPADs between 4.5 and 9hpf of zebrafish embryogenesis. Ranking of MPADs is the same as in *Figure 4F*. (B) Average change in H3K27ac signal between 4.5hpf and 9hpf (log2 of fold-change) at MPADs. (C, D) Box plots showing expression of genes associated (≤30 kb) with each of the four MPAD classes, as determined by RNA-seq on 6hpf and 12hpf embryos. Data are presented as log2 of mean TPM values at 12hpf (C) or as log2 of mean TPM fold-change between 12hpf and 6hpf (D). Statistical test: pairwise comparison with Kruskal-Wallis followed by Dunn's post-hoc test.

DOI: https://doi.org/10.7554/eLife.36144.011

The following figure supplement is available for figure 5:

**Figure supplement 1.** Non-MPADs undergo changes in chromatin state during embryogenesis.

DOI: https://doi.org/10.7554/eLife.36144.012

MPADs undergo a reduction in the level of H3K27ac modification from 4.5hpf to 9hpf (*Figure 5A,B*), while RNA-seq at 12hpf (to capture changes in gene expression corresponding to chromatin changes at 9hpf; *Figure 5C*) shows that the associated genes are expressed at similar levels at 12hpf and 6hpf (*Figure 5D*). In contrast, Class 4 MPADs display higher levels of H3K27ac at 9hpf than at 4.5hpf and their associated genes show the greatest increase in expression between 6hpf and 12hpf. Class 3 MPADs show an intermediate effect with a small change in H3K27ac levels and a slight increase in expression of associated genes. We also find that many of the TALE-occupied regions that are sparsely modified at 4.5hpf (non-MPADs defined in *Figure 4A*) become more highly modified by H3K27ac as development progresses (*Figure 5—figure supplement 1A,B*). Genes associated with the non-MPADs undergoing the greatest increase in H3K27ac levels show the greatest increase in expression (*Figure 5—figure supplement 1C*) and are also enriched for functions related to later stages of embryogenesis (*Figure 5—figure supplement 1D*). Hence, Class 4 MPADs (and, to a lesser extent, Class 3 MPADs and non-MPADs) undergo an increase in H3K27ac and expression of the associated genes is significantly upregulated by 12hpf.

## TALE factors control the chromatin state at class 4 MPADs associated with the anterior GRN

The fact that developmental control genes are associated with Class 4 MPADs suggests that the TALE GRN genes may fall into this category. Indeed, we find that TALE GRN genes are significantly associated with Class 4 (and Class 3), but not Class 1 or 2, MPADs (*Figure 6A,B*). A closer analysis of the TALE GRN genes associated with Class 3 and 4 MPADs revealed that they are enriched for functions related to transcriptional regulation and early embryonic processes (*Figure 6C*) that align well with the developmental defects observed in TALE KD embryos. In fact, 27 of the 34 TALE GRN genes associated with Class 4 MPADs encode TFs (*Figure 6E*) and a literature review uncovered that ~65% (22/34) have been previously implicated in the formation of embryonic structures that are affected in TALE KD embryos (*Figure 6E*; *Supplementary file 4*). These findings suggest that TALE factors act via Class 4 (and, to a certain extent, Class 3) MPADs to control a core set of TFs in the TALE GRN. To directly test this possibility, we assessed whether TALE factors are required for the expression of MPAD-associated genes by 12hpf. We find that expression of genes associated with Class 1 and 2 MPADs is relatively insensitive to TALE KD, while genes associated with Class three and, in particular, Class 4 MPADs are downregulated in TALE KD embryos (*Figure 6D,E*). Since Class 4 MPADs show an increase in H3K27ac between 6hpf and 9hpf (*Figure 5A*), we examined the impact of TALE TFs on 9hpf H3K27ac levels. Using ChIP-qPCR, we find that H3K27ac levels are reduced at 57% (4/7) of TALE GRN-associated Class 4 MPADs in TALE KD embryos (*Figure 6F*). These findings indicate that TALE factors act by regulating a chromatin transition – from repressive chromatin in blastula stage embryos to active chromatin in segmentation stage embryos – at a core set of genes encoding TFs that direct primarily anterior development in the zebrafish embryo.

## NF-Y proteins regulate TALE GRN expression and form complexes with TALE factors

Since TALE factors commonly function in complexes with other TFs, it is possible that they have novel interaction partners when bound at DECA motifs. Indeed, the DREME discovery tool detected three motifs in addition to the DECA motif at $Prep_{3.5hpf}$ peaks (*Figure 7A*). We cannot confidently assign a TF to the AT(A/G)TTAA motif, and the CC(C/A)C(G/A)CCC motif could bind any member of the large Sp/Klf family. The CCAAT motif was detected in a previous Prep ChIP-seq analysis (*Penkov et al., 2013*), but it was not pursued further. In our analysis, DREME predicted this motif to be selective for the NF-Y transcription factor (*Dolfini et al., 2009*). While the other motifs are enriched at both $Prep_{3.5hpf}$ and $Prep_{12hpf-only}$ peaks, the NF-Y motif is specifically enriched at $Prep_{3.5hpf}$ peaks (*Figure 7B*). NF-Y is also maternally deposited in zebrafish (*Figure 7—figure supplement 1A*), consistent with a joint role for TALE and NF-Y factors at blastula stages. Using ChIP-qPCR, we tested 15 TALE-occupied sites with nearby CCAAT motifs and detect NF-Y binding at nine of them (*Figure 7C*), demonstrating that co-occupancy is relatively frequent. Accordingly, using ChIP-seq data from mESCs (*Oldfield et al., 2014*), we find that ~50% of all Prep peaks are found near NF-Y peaks also in this cell type (*Figure 7D*), demonstrating that co-localization of TALE and NF-Y TFs is evolutionarily conserved.

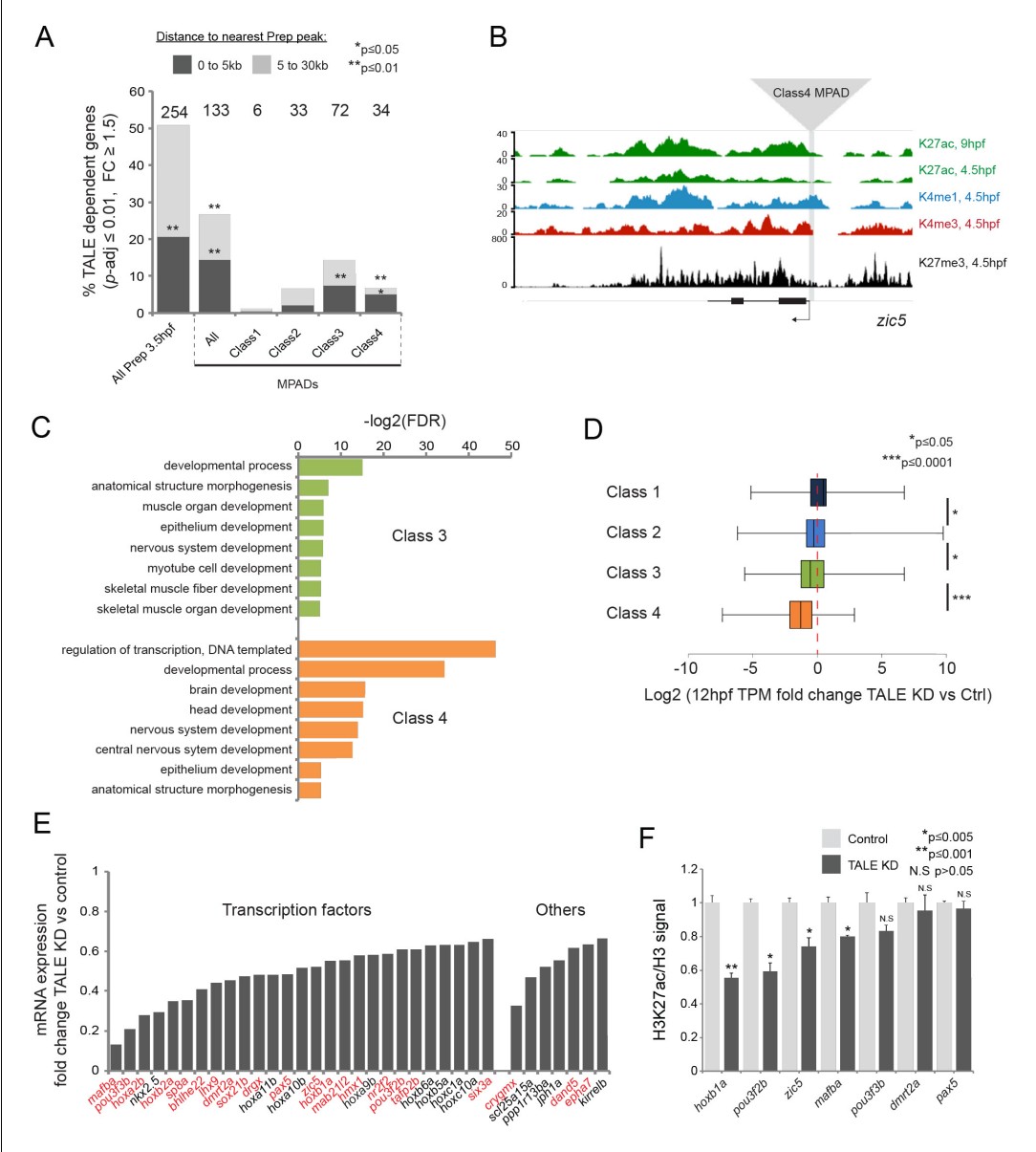

**Figure 6.** TALE factors control the chromatin state at Class 4 MPADs associated with the TALE GRN. (**A**) Localization of TALE KD downregulated genes (p-adj ≤0.01, fold-change ≥1.5) relative to MPADs. The number of TALE-dependent genes within 30 kb of MPADs is indicated above each bar. *p*-values for enrichment above a random set of genes were calculated using the Pearson correlation test. (**B**) Representative UCSC browser tracks of the *zic5* locus illustrating the position of a Class 4 MPAD and histone modifications in 4.5hpf and 9hpf embryos. (**C**) DAVID analysis of TALE KD downregulated genes (p-adj ≤0.01, fold-change ≥1.5) near Class 3 and 4 MPADs. Note that only select categories are presented, a full list of GO terms is available in ***Supplementary file 3***. FDR = Benjamini multiple testing False Discovery Rate. (**D**) Box plots showing change in expression of genes near (≤30 kb) each of the four MPAD classes, as determined by RNA-seq at 12hpf. Data are presented as log2 of mean TPM fold-change between TALE KD and control. Statistical test: pairwise comparison with Kruskal-Wallis followed by Dunn's post-hoc test. (**E**) Graph showing the TPM expression fold-change in TALE KD vs control 12hpf embryos for all TALE dependent genes (n = 34) near (≤30 kb) Class 4 MPADs. Genes in red control the formation of structures affected by TALE KD (see ***Supplementary file 4***). (**F**) H3K27ac/Histone H3 signal ratio at Class 4 MPADs as determined by ChIP-qPCR in 9hpf control vs TALE KD embryos. MPADs are labeled with the name of the nearest TALE-dependent gene. Data of three independent biological replicates are presented as mean fold change ± SEM of TALE KD vs control. Statistical test: unpaired t-test.

DOI: https://doi.org/10.7554/eLife.36144.013

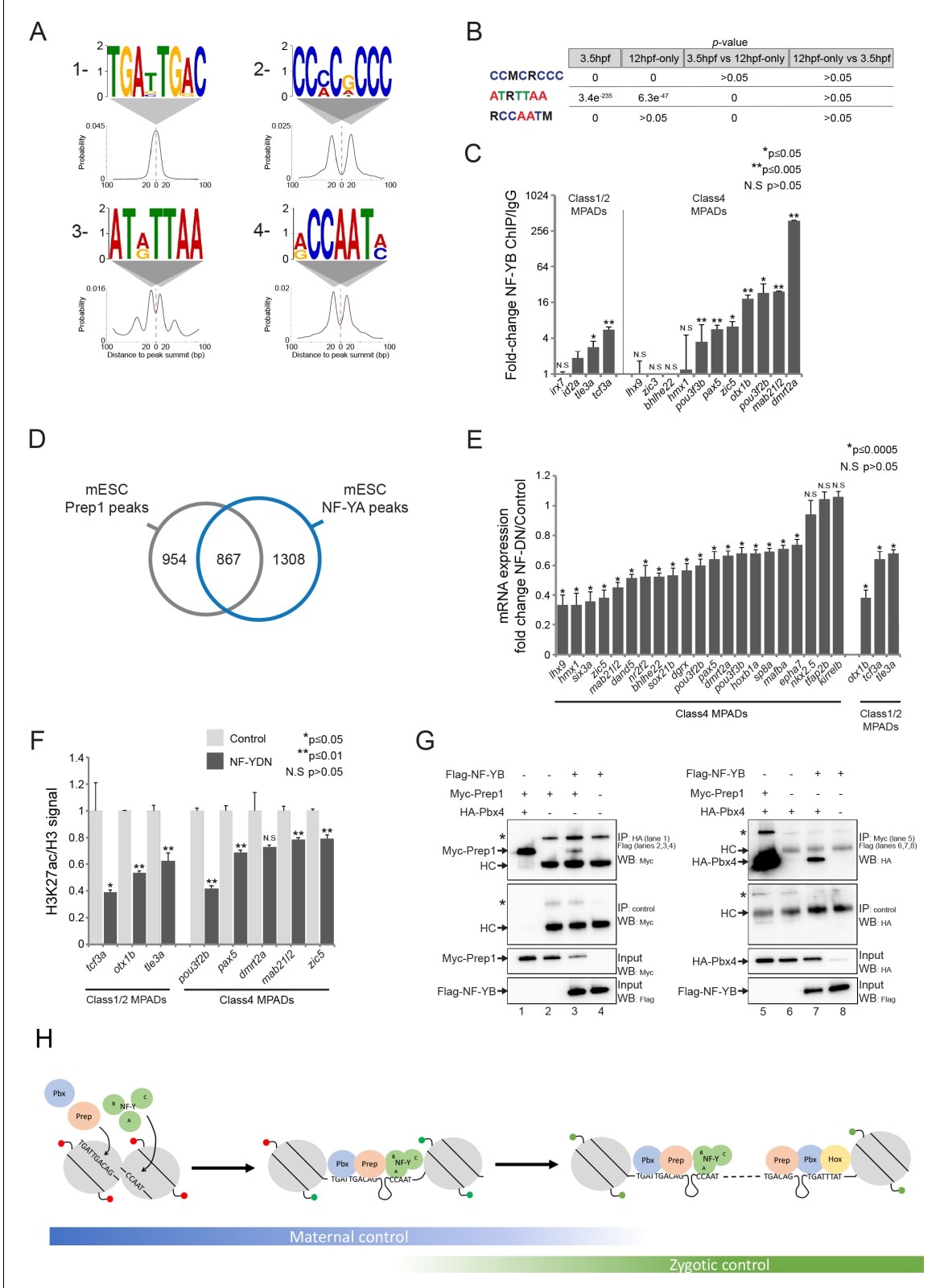

**Figure 7.** NF-Y proteins regulate TALE GRN expression and form complexes with TALE factors. See also *Figure 7—figure supplement 1*. (**A**) Sequence logo and localization relative to Prep peak summits of motifs identified by DREME at Prep_{3.5hpf} peaks. (**B**) Enrichment of motifs in Prep_{3.5hpf} and Prep_{12hpf-only} peaks as defined by AME. *p*-values for enrichment above random occurrence (3.5hpf and 12hpf-only columns) or between two Prep peak populations (3.5hpf vs 12hpf-only and 12hpf-only vs 3.5hpf columns) were calculated using the ranksum test in AME. Motifs are represented in

*Figure 7 continued*

IUPAC code (M = A or C; R = A or G). (**C**) ChIP-qPCR showing NF-YB binding at CCAAT motif-containing MPADs in 9hpf embryos. MPADs are labeled with the name of the nearest gene. Data of three independent biological replicates are presented as mean fold change ± SEM of NF-YB IP vs control IgG. Statistical test: unpaired t-test. (**D**) Venn diagram illustrating the overlap of Prep and NF-YB peaks in mESCs. Two peaks are considered to overlap if their summits are within 500 bp. (**E**) RT-qPCR analysis of gene expression in 12hpf NF-YDN injected embryos. Results are shown as gene expression fold-change in NF-YDN vs control for select TALE-dependent genes. Data of three independent experiments are presented as mean fold change ± SEM of NF-YDN injected vs control embryos. Statistical test = unpaired t-test. (**F**) H3K27ac/Histone H3 signal ratio at MPADs (labeled with the name of the nearest gene) as determined by ChIP-qPCR in 9hpf control vs NF-YDN injected embryos. Data of three independent biological replicates are presented as mean fold change ± SEM of NF-YDN vs control. Statistical test: unpaired t-test. (**G**) Co-IP experiments showing interaction of Myc-Prep (left panels) and HA-Pbx4 (right panels) with Flag-NF-YB in transfected HEK293 cells. HC = Ig heavy chain. Asterisks indicate non-specific signal. (**H**) Model diagram. At blastula stages (left side) TALE binds DECA motifs (TGATTGACAG) near NF-Y motifs (CCAAT). At this stage, most binding sites are occupied by nucleosomes and those associated with developmental control genes are marked by H3K27me3 (red lollipops). Binding of TALE and NF-Y leads to deposition of H3K27ac (green lollipops) and improved accessibility. At segmentation stages (right side), TALE continues to bind DECA motifs near NF-Y motifs, but Prep also binds HEXA motifs (TGACAG) near PBX:HOX motifs (TGATTTAT). Most of the HEXA motifs lack nucleosomes and are found within 40 kb of a DECA/NF-Y site (indicated by dashed connecting line). At this stage, developmental control genes are marked by H3K27ac and are expressed.

DOI: https://doi.org/10.7554/eLife.36144.014

The following figure supplement is available for figure 7:

**Figure supplement 1.** NF-Y TF regulates anterior embryonic structures and interacts with Prep and Pbx.

DOI: https://doi.org/10.7554/eLife.36144.015

The role for NF-Y in embryogenesis is not well characterized, but it has been reported that mice mutant for *nf-ya* (the DNA binding subunit of the NF-Y complex) die *in utero* prior to embryonic day 8.5 (*Bhattacharya et al., 2003*), consistent with a role for NF-Y in early embryogenesis. Furthermore, a study targeting zebrafish *nf-yb* with antisense morpholino oligos described a relatively mild head phenotype that was attributed to defective cartilage formation (Y.-H. *Chen et al., 2009*). Using a previously reported dominant negative construct (NF-YDN [*Nardini et al., 2013*; *Mantovani et al., 1994*]) to disrupt NF-Y function, we observe a small head, as well as defects in development of the eyes, heart and tail (*Figure 7—figure supplement 1B*). The effect of the NF-YDN is somewhat more severe than that resulting from TALE KD (*Figure 1—figure supplement 1A,B*), but the two phenotypes share some features – including smaller head and eyes, as well as cardiac edema – suggesting that NF-Y may also regulate the expression of genes in the TALE GRN. To test this, we analyzed expression of 21 TALE-dependent genes associated with Class 4 MPADs (out of the 34 such genes identified in *Figure 6A*; six of these were also confirmed as associated with NF-Y occupancy in *Figure 7C*) and find that 18 (86%) are downregulated upon NF-Y disruption (*Figure 7E*). Furthermore, NF-Y disruption leads to a decrease in H3K27ac at MPADs associated with these genes (*Figure 7F*), similar to our observation following disruption of TALE function (*Figure 6F*). A shared role for TALE and NF-Y factors in controlling H3K27ac may be broadly relevant at the blastula stage, since we find that TALE peaks with adjacent CCAAT motifs are generally associated with higher levels of H3K27ac and lower levels of H3K27me3 than TALE peaks that lack a nearby CCAAT box (*Figure 7—figure supplement 1C*). We do not find any differences in the distribution of NF-Y motifs among the various MPAD classes, suggesting that NF-Y is generally associated with TALE occupancy (*Figure 7—figure supplement 1D*). We noticed from our bioinformatics analysis that NF-Y sites occur very close to DECA sites, with the average spacing being ~20 bp (*Figure 7A*), raising the possibility that NF-Y may physically interact with TALE proteins. Since Prep:Pbx is a heterodimer and NF-Y is a heterotrimeric TF, we tested the ability of Prep and Pbx to bind NF-YA and/or NF-YB in pairwise combinations by co-immunoprecipitation from transfected HEK293 cells. In this context, we find that both Prep and Pbx interact with the NF-YB (*Figure 7G*) and NF-YA (*Figure 7—figure supplement 1F*) subunits, indicating that Prep:Pbx and NF-Y can form complexes. We conclude that NF-Y binds adjacent to TALE factors at DECA sites and that both factors are required for regulation of the TALE GRN, possibly by functioning in a complex.

As discussed above, genomic elements containing HEXA and PBX:HOX motifs have been shown to function as enhancers (*Pöpperl et al., 1995*; *Jacobs et al., 1999*; *Ferretti et al., 2005*; *Choe et al., 2009*; *Ferretti et al., 2000*; *Di Rocco et al., 1997*; *Manzanares et al., 2001*; *Tümpel et al., 2007*; *Wassef et al., 2008*), but it is not clear if elements containing DECA and NF-Y sites have such activity. In particular, most TALE GRN genes associated with Class 4 MPADs have

tissue-specific expression patterns, but the TALE and NF-Y factors are ubiquitously expressed, suggesting that genomic elements containing only DECA and NF-Y sites may not be sufficient to drive gene expression. Accordingly, by testing seven DECA and NF-Y site-containing genomic elements for enhancer activity in HEK293 cells, we find that only one drives luciferase reporter expression (*Figure 7—figure supplement 1E*). This finding is consistent with previous reports that mis-expression of TALE factors in zebrafish embryos does not cause developmental defects (*Vlachakis et al., 2001*; *Choe et al., 2002*) and suggests that elements containing DECA and NF-Y sites function together with other regulatory elements that provide tissue-specific input (see Discussion).

## Discussion

In its previously defined role as acting in complexes with Hox TFs, Prep binds at monomeric HEXA sites near binding sites for Pbx:Hox dimers to control gene expression (*Ferretti et al., 2005*; *Tümpel et al., 2007*; *Jacobs et al., 1999*; *Ferretti et al., 2000*; *Amin et al., 2015*). Accordingly, our analysis detected HEXA motifs with nearby PBX:HOX motifs at Prep binding sites associated with a TALE-dependent anterior GRN in segmentation stage (12hpf) zebrafish embryos. Strikingly, we find that TALE-occupancy is associated with this GRN already at blastula stages (3.5hpf), but at this stage TALE factors instead utilize DECA sites (consisting of immediately adjacent Pbx and Prep sites). We also discovered that NF-Y binds CCAAT motifs near DECA sites and forms complexes with TALE factors. Finally, we demonstrate that TALE and NF-Y are both required for the transition to an active chromatin profile at GRN-associated genes. Hence, TALE factors control an anterior GRN throughout embryogenesis, but the choice of binding motifs and partner proteins varies such that TALE factors interact with NF-Y at DECA sites starting at blastula stages and then expand their binding repertoire to also include HEXA sites, where they interact with Pbx:Hox dimers, by segmentation stages (see summary model in *Figure 7H*).

Although DECA sites were identified previously (*Penkov et al., 2013*; *De Kumar et al., 2017*; *Laurent et al., 2015*; *Knoepfler and Kamps, 1997*; *Chang et al., 1997*), they have not been assigned a biological function. Our experiments now reveal that genomic elements containing DECA and NF-Y motifs may not be sufficient to act as enhancers. Instead, TALE and NF-Y bind many of these elements prior to the appearance of active chromatin marks. Indeed, we note that many genomic loci bound by Prep at 3.5hpf are highly occupied by nucleosomes (*Figures 3E* and *4A*), indicating that Prep can access its binding sites in compacted embryonic chromatin. Furthermore, we find that TALE factors are required for the deposition of H3K27ac marks at these elements (*Figure 6F*). This may be a general function of TALE factors since several TALE proteins bind CBP (*Choe et al., 2009*; *Saleh et al., 2000*) – the enzyme responsible for H3K27 acetylation (*Tie et al., 2009*) – and Pbx reportedly promotes active chromatin in a breast cancer cell line (*Magnani et al., 2011*). Additionally, NF-Y contains a histone-fold and makes both specific and non-specific contacts with DNA (*Nardini et al., 2013*), suggesting that NF-Y may access its binding site by displacing histones. Hence, the joint activity of TALE and NF-Y may represent a pioneer function (*Iwafuchi-Doi and Zaret, 2016*) that permits access to DECA/NF-Y sites in compacted chromatin (see summary model in *Figure 7H*). Although only ~50% of TALE-occupied sites are associated with a NF-Y motif at 3.5hpf, there are also nearby motifs for SP/KLF (*Figure 7A*) and KLF4 is a pioneer factor (*Soufi et al., 2015*) that binds TALE proteins (*Bjerke et al., 2011*), suggesting that TALE proteins may act together with various other TFs in a pioneer role at DECA sites.

We find that many of the TALE-dependent genes identified by our analysis are expressed in the anterior embryo. Since TALE and NF-Y factors are present ubiquitously, this suggests that additional tissue-restricted inputs are required to achieve spatially appropriate expression of these genes during embryogenesis. We therefore hypothesize that TALE and NF-Y pioneer activity is required for nearby tissue-specific enhancers to become functional (see summary model in *Figure 7H*). In fact, the additional Prep-occupied sites that emerge by 12hpf may represent such tissue-specific enhancers. Some of these sites contain monomeric HEXA motifs near PBX:HOX motifs in an arrangement found at many hindbrain enhancers (*Grice et al., 2015*) and they are enriched near DECA/NF-Y sites. These 12hpf Prep sites contain not only PBX:HOX binding sites, but also motifs for other tissue-specific TFs (such as myogenic factors) indicating that DECA/NF-Y motifs may play a general role in promoting access to enhancers. We also note that TALE factors arose prior to Hox genes in evolution (*Bürglin and Affolter, 2016*; *Hrycaj and Wellik, 2016*; *Holland, 2013*), suggesting that

TALE activity at DECA sites may represent an original function and that TALE factors may have been subsequently co-opted to function together with tissue-specific TFs.

Maternally deposited material controls embryonic development in zebrafish until 3hpf-4hpf. Indeed, TALE and NF-Y are maternally deposited in zebrafish ([*Deflorian et al., 2004*; *Choe et al., 2002*; *Waskiewicz et al., 2002*; *Chen et al., 2009*]; *Figure 2—figure supplement 1A,B*; *Figure 7— figure supplement 1A*) and by 3.5hpf – the stage when we carried out our ChIP-seq analysis – zygotic Prep, Pbx and NF-Y expression is not yet detectable (*Figure 2—figure supplement 1A*, *Figure 7—figure supplement 1A*). Hence, the initial activity of TALE and NF-Y at DECA/NF-Y sites at 3.5hpf is likely maternally directed, while DECA/NF-Y sites and HEXA/PBX:HOX sites detected at 12hpf are more likely occupied by zygotically produced factors. Differences between maternally and zygotically controlled stages of embryogenesis may also explain why Prep binds HEXA sites efficiently at 12hpf, but not at 3.5hpf. Specifically, it is possible that Prep cannot bind HEXA sites as a monomer but requires the cooperation of tissue-specific TFs (such as Hox proteins) that are not present maternally. Indeed, our recent work demonstrated that binding of Meis proteins (that are closely related to Prep proteins) to HEXA motifs is stabilized by Hox proteins in segmentation stage mouse embryos (*Amin et al., 2015*).

Prep binds many genomic loci in the 3.5hpf embryo and these sites display diverse chromatin states, such that Class 1 and 2 MPADs are associated with genes expressed by 6hpf, Class 4 MPADs with genes expressed by 12hpf and non-MPADs with genes expressed at later stages of embryogenesis (*Figures 4F–G* and *5C–D*, *Figure 5—figure supplement 1*). While our functional analysis indicates that primarily genes associated with Class 4 MPADs are affected by TALE KD (*Figure 6D*), this is likely a result of our choosing the 12hpf timepoint for RNA-seq. Indeed, we show that non-MPADs continue to transition to an active chromatin state at least until 24hpf (*Figure 5—figure supplement 1*), but any genes that become expressed as a result of this transition would not have been detected by our analysis. For instance, muscle differentiation involves TALE function (*Berkes et al., 2004*; *Knoepfler et al., 1999*) and Prep peaks are found near genes involved in myogenesis (*Figure 2E*). Although expression of myogenic genes is somewhat affected in TALE KD embryos (*Figure 1D,E*) much of muscle differentiation takes place after 12hpf suggesting that this expression effect would be more pronounced at later stages. Accordingly, the effect of TALE factors at Class 3 MPAD-associated genes is less pronounced (*Figure 6D*), possibly because these genes are involved in muscle development (*Figure 6C*). Genes associated with Class 1 and 2 MPADs are only mildly TALE-dependent (*Figure 6D*). Strikingly, ~70% of 'first-wave' genes (ones activated by maternal factors in the early zygote [*Lee et al., 2013*]) are located near Prep peaks (*Figure 7—figure supplement 1G*) – particularly near Class 1 and 2 MPADs (*Figure 7—figure supplement 1H*) – but expression of these genes is not affected by TALE KD (*Figure 1—figure supplement 1D–F*). The reason for this is not clear, but the pluripotency factors Nanog, Pou5fl and SoxB1 are required for expression of first-wave genes (*Leichsenring et al., 2013*; *Lee et al., 2013*) and may act redundantly with TALE and NF-Y at these early stages. Accordingly, our RNA-seq analysis found that expression of *nanog*, *pou5fl* and *soxB1* is not disrupted in TALE KD embryos. Alternatively, the onset of the knockdown effect may be delayed, preventing it from disrupting early TALE activity required for first-wave gene expression.

Lastly, TALE factors act as oncogenes in several systems and have been specifically implicated in various types of leukemia (*Kamps and Baltimore, 1993*; *Nourse et al., 1990*; *Moskow et al., 1995*). Their oncogenic potential has generally been considered in the context of their action as transcription cofactors to Hox proteins (*Eklund, 2007*). Our finding that TALE factors use additional binding motifs and interaction partners, as well as their ability to promote an active chromatin state, suggests that this model should be expanded to also consider non Hox-related mechanisms for TALE factor-mediated leukemogenesis.

# Materials and methods

**Key resources table**

| Reagent type or resources | Designation | Source or reference | Identifier | Additional information |
|---|---|---|---|---|

*Continued on next page*

Continued

| Reagent type or resources | Designation | Source or reference | Identifier | Additional information |
|---|---|---|---|---|
| Antibody | Rabbit polyclonal anti-Prep | (*Choe et al., 2014*) | N/A | |
| Antibody | Rabbit polyclonal anti-Pbx4 | (*Choe et al., 2014*) | N/A | |
| Antibody | Rabbit polyclonal anti-NF-YB | Santa-Cruz | sc13045 | RRID:AB_2152107 |
| Antibody | Rabbit polyclonal anti-H3K27ac | Abcam | ab4729 | RRID:AB_2118291 |
| Antibody | Rabbit polyclonal anti-Histone H3 | Abcam | ab1791 | RRID:AB_302613 |
| Antibody | Mouse monoclonal anti-Myc | Roche | 11667149001 | RRID:AB_390912 |
| Antibody | Mouse monoclonal anti-Flag | Sigma-Aldrich | F3165 | RRID:AB_259529 |
| Antibody | Rabbit polyclonal anti-HA | Abcam | ab9110 | RRID:AB_307019 |
| Antibody | Rabbit polyclonal anti-IgG control | Abcam | ab46540 | RRID:AB_2614925 |
| Antibody | Mouse polyclonal anti-IgG control | Millipore | 12-371b | RRID:AB_2617156 |
| Antibody | Anti-mouse IgG, HRP-linked secondary antibody | GE healthcare | LNA91V/AG | |
| Antibody | Anti-mouse IgG, Alexa Fluor 488 conjugated secondary antibody | Molecular Probes | A11001 | RRID:AB_2534069 |
| Antibody | Mouse monoclonal 3A10 | Developmental Studies Hybridoma bank | 531874 | RRID:AB_531874 |
| Antibody | Anti-rabbit IgG, HRP-linked secondary antibody | Jackson Laboratories | 211-032-171 | RRID:AB_2339149 |
| Antibody | Lipofectamine 2000 | Invitrogen | 52887 | |
| Strain, strain background (*E. coli*) | Subcloning Efficiency DH5α Competent Cells | ThermoFisher Scientific | 18265017 | |
| Chemical compound, drug | 4-Thiouridine | Santa-Cruz | sc204628 | |
| Chemical compound, drug | EZ-Link HPDP-Biotin | Pierce | 21341 | |
| peptide, recombinant protein | Dynabeads MyOne Streptavidin C1 | ThermoFisher Scientific | 65001 | |
| peptide, recombinant protein | Protein-A Dynabeads | ThermoFisher Scientific | 10001D | |
| Commercial assay or kit | TruSeq ChIP Library Preparation Kit | Illumina | IP-202–1012 | |
| Commercial assay or kit | TruSeq Stranded mRNA LT sample prep Kit | Illumina | RS-122–2101 | |
| Commercial assay or kit | mMESSAGE mMACHINE SP6 Transcription Kit | ThermoFisher Scientific | AM1340 | |
| Commercial assay or kit | Q5 Site-Directed Mutagenesis Kit | New England Biolabs | E0554S | |
| Other | Prep ChIP-seq and Inputs in 3.5hpf zebrafish embryos | This paper | GEO | Deposited data |

*Continued on next page*

*Continued*

| Reagent type or resources | Designation | Source or reference | Identifier | Additional information |
|---|---|---|---|---|
| Other | Prep ChIP-seq and Inputs in 12hpf zebrafish embryos | This paper | GEO | Deposited data |
| Other | TALE knock-down and control RNA-seq in 6hpf zebrafish embryos | This paper | GSE102662 | Deposited data |
| Other | TALE knock-down and control RNA-seq in 12hpf zebrafish embryos | This paper | GSE102662 | Deposited data |
| Other | Prep1 ChIP-seq and Inputs in mESCs, WIG files | (*Laurent et al., 2015*) | GSM1545025 and GSM1545026 | Deposited data |
| Other | ATAC-seq in 4hpf zebrafish embryos, fastq files | (*Kaaij et al., 2016*) | SRR2747531 | Deposited data |
| Other | H3K4me1 ChIP-seq in 4.5hpf zebrafish embryos, WIG files | (*Bogdanovic et al., 2012*) | GSM915193 | Deposited data |
| Other | H3K4me3 ChIP-seq in 4.5hpf zebrafish embryos, WIG files | (*Bogdanovic et al., 2012*) | GSM915189 | Deposited data |
| Other | H3K27ac ChIP-seq in 4.5hpf zebrafish embryos, WIG files | (*Bogdanovic et al., 2012*) | GSM915197 | Deposited data |
| Other | H3K27ac ChIP-seq in 9hpf zebrafish embryos, WIG files | (*Bogdanovic et al., 2012*) | GSM915198 | Deposited data |
| Other | H3K27ac ChIP-seq in 24hpf zebrafish embryos, WIG files | (*Bogdanovic et al., 2012*) | GSM915199 | Deposited data |
| Other | H3K27me3 ChIP-seq in 4.5hpf zebrafish embryos, WIG files | (*Zhang et al., 2014*) | GSM1081557 | Deposited data |
| Other | MNase-seq in 4.5hpf zebrafish embryos, WIG files | (*Zhang et al., 2014*) | GSM1081554 | Deposited data |
| Other | RNA-Pol2 ChIP-seq in 4.5hpf zebrafish embryos, WIG files | (*Zhang et al., 2014*) | GSM1081560 | Deposited data |
| Other | MeDIP-seq (Methyl CpG) in 4.5hpf zebrafish embryos, BedGraph files | (*Lee et al., 2015*) | GSM1274386 | Deposited data |
| Other | NF-YA ChIP-seq in mESCs | (*Oldfield et al., 2014*) | GSM1370111 | Deposited data |
| Other | H3K4me1 in mESCs, BigWig files | ENCODE www.encodeproject.org | GSM1000121 | Deposited data |
| Other | H3K4me3 in mESCs, BigWig files | ENCODE www.encodeproject.org | GSM1000124 | Deposited data |
| Other | H3K27ac in mESCs, BigWig files | ENCODE www.encodeproject.org | GSM1000126 | Deposited data |
| Other | H3K27me3 in mESCs, BigWig files | ENCODE www.encodeproject.org | GSM1000089 | Deposited data |
| Other | DNase-seq in mESCs, BigWig files | ENCODE www.encodeproject.org | GSM1014154 | Deposited data |
| Other | MeDIP-seq (Methyl CpG) in mESCs | (C.-C. *Chen et al., 2013*) | GSM859494 | Deposited data |
| Cell line (Human) | HEK-293T cells | ATCC | ATCC CRL-3216 | RRID:CVCL_0063 |
| Strain, strain background (Zebrafish) | strain EKW | Ekkwill breeders | http://www.ekkwill.com/ | |
| Other | Oligonucleotides | | | See *Supplementary file 5* |
| Recombinant DNA | 6xMyc-Prep1.1 in PCS2 + MT | (*Choe et al., 2002*) | N/A | |

*Continued on next page*

Continued

| Reagent type or resources | Designation | Source or reference | Identifier | Additional information |
|---|---|---|---|---|
| Recombinant DNA | HA-Pbx4 in PCS2+ | (*Choe et al., 2009*) | N/A | |
| Recombinant DNA | Flag-NF-YA in PCS2+ | This Paper | N/A | |
| Recombinant DNA | Flag-NF-YB in PCS2+ | This Paper | N/A | |
| Recombinant DNA | NF-YDN in PCS2+ | This paper | N/A | |
| Recombinant DNA | pGL3-Promoter vector | Promega | E1761 | |
| Recombinant DNA | Tle3 element in pGL3 Promoter vector | This paper | N/A | |
| Recombinant DNA | Pax5 element in pGL3 Promoter vector | This paper | N/A | |
| Recombinant DNA | Prdm14 element in pGL3 Promoter vector | This paper | N/A | |
| Recombinant DNA | Tcf3a element in pGL3 Promoter vector | This paper | N/A | |
| Recombinant DNA | Her6 element in pGL3 Promoter vector | This paper | N/A | |
| Recombinant DNA | Dachb element in pGL3 Promoter vector | This paper | N/A | |
| Recombinant DNA | Fgf8 element in pGL3 Promoter vector | This paper | N/A | |
| Recombinant DNA | pGL3-Control vector | Promega | E1741 | |
| Software, algorithm | FastQC | Babraham Institute | https://www.bioinformatics.babraham.ac.uk/projects/fastqc/ | RRID:SCR_014583 |
| Software, algorithm | FastQ Screen | Babraham Institute | https://www.bioinformatics.babraham.ac.uk/projects/fastq_screen/ | RRID:SCR_000141 |
| Software, algorithm | Trimmomatic 0.32 | (*Bolger et al., 2014*) | https://github.com/timflutre/trimmomatic | RRID:SCR_011848 |
| Software, algorithm | Bowtie 2.2.3 | (*Langmead and Salzberg, 2012*) | https://github.com/BenLangmead/bowtie2 | RRID:SCR_005476 |
| Software, algorithm | SAMtools 0.1.19 | (*Li et al., 2009*) | https://github.com/samtools/samtools | RRID:SCR_002105 |
| Software, algorithm | MACS 2.1.0.20140616 | (*Zhang et al., 2008*) | https://github.com/taoliu/MACS | |
| Software, algorithm | RSEM 1.2.28 in the Dolphin interface of University of Massachuetts Worcester Biocore | (*Li and Dewey, 2011*) | http://www.umassmed.edu/biocore/introducing-dolphin/ | RRID:SCR_013027 |
| Software, algorithm | DESeq2 in the Dolphin interface of University of Massachuetts Worcester Biocore | (*Anders and Huber, 2010*) | http://www.umassmed.edu/biocore/introducing-dolphin/ | RRID:SCR_015687 |
| Software, algorithm | Galaxy web interface | (*Goecks et al., 2010*) | https://usegalaxy.org | RRID:SCR_006281 |
| Software, algorithm | BedTools in galaxy | (*Quinlan and Hall, 2010*) | https://usegalaxy.org | RRID:SCR_006646 |
| Software, algorithm | DeepTools in galaxy | (*Ramírez et al., 2014*) | https://usegalaxy.org | |
| Software, algorithm | MEME-ChIP | (*Machanick and Bailey, 2011*; *Bailey et al., 2009*) | http://meme-suite.org/tools/meme-chip | RRID:SCR_001783 |
| Software, algorithm | DAVID 6.8 | (*Huang et al., 2009b*, *2009a*) | https://david.ncifcrf.gov/ | RRID:SCR_001881 |

*Continued on next page*

Continued

| Reagent type or resources | Designation | Source or reference | Identifier | Additional information |
|---|---|---|---|---|
| Software, algorithm | GREAT 3.0.0 | (*McLean et al., 2010*; *Hiller et al., 2013*) | http://bejerano.stanford.edu/great/public/html | RRID:SCR_005807 |
| Other | anti-Prep1.1 morpholino oligonucleotide | Gene Tools, LLC | N/A | |
| Other | 5'-TGGACACAGACTGGGCAGCCATCAT-3'Fluorescein | (*Deflorian et al., 2004*) | | |
| Other | anti-Pbx2 morpholino oligonucleotide | Gene Tools, LLC | N/A | |
| Other | 5'-CCGTTGCCTGTGATGGGCTGCTGCG-3' | (*Erickson et al., 2007*) | | |
| Other | anti-Pbx4 morpholino oligonucleotide | Gene Tools, LLC | N/A | |
| Other | 5'-AATACTTTTGAGCCGAATCTCTCCG-3' | (*Erickson et al., 2007*) | | |

## Animal care

All procedures on zebrafish adults and embryos were approved by the University of Massachusetts Institutional Animal Care and Use Committee (IACUC). EKW zebrafish were kept in groups of 10 individuals under constant water flow at 28°C. To collect embryos, 2 males and three females were crossed for 30 min. Subsequently, the embryos were collected in egg water (60 ug/ml of instant ocean salts, 0.0002% methylene blue). After 2 hr, dead and un-fertilized embryos were manually removed and the remainder left to develop until they reached the appropriate developmental stage and then used in the experimental procedures described below.

## Interference with protein function in embryos

Injection of capped messenger RNAs encoding an NF-Y or a Prep/Meis dominant negative protein (NF-YDN and PBCAB, respectively [*Mantovani et al., 1994*; *Choe et al., 2002*]) or a cocktail of morpholino antisense oligonucleotides directed against the TALE proteins, were used to interfere with NF-Y and TALE function. TALE knockdown was achieved by injection of antisense morpholino oligos (MOs) targeting *pbx2*, *pbx4* and *prep1.1* as reported previously (*Deflorian et al., 2004*; *Waskiewicz et al., 2002*). The use of MOs is necessitated by the fact that mutant lines are not available for all TALE factors, and the existing mutants are embryonic lethal. Hence, MOs allow us to produce the large number of embryos required for RNA-seq and ChIP-qPCR experiments. Importantly, the phenotype of *pbx4* MO-injected embryos is indistinguishable from that of *pbx4* mutant embryos (*Waskiewicz et al., 2002*), demonstrating that *pbx4* MOs are specific. *prep1.1* MOs produce the same phenotype as *pbx4* mutants (*Deflorian et al., 2004*), as expected of proteins acting together in a dimer. *prep1.1* MOs also produce the same phenotype as embryos injected with a dominant negative construct disrupting Prep/Meis function (*Choe et al., 2002*), further indicating that the knockdown is specific.

Sample size was not selected based on statistical analysis, but on previous published reports demonstrating that these reagents produce phenotypes in >85% of injected embryos (*Deflorian et al., 2004*; *Waskiewicz et al., 2002*; *Choe et al., 2014*; *Mantovani et al., 1994*). Embryos were randomly selected for inclusion in injected or control pools. Dead animals were excluded from RNA-seq and ChIP-seq experiments, but not from phenotypic analyses in *Figure 1—figure supplement 1* and *Figure 7—figure supplement 1*. No other animals were excluded. Experiments were not blinded.

## In vitro synthesis of capped mRNAs

PCS2 + plasmids containing the NF-YDN or PBCAB coding sequence was linearized by NotI digest and purified with a PCR purification kit column (Qiagen). Capped messenger RNAs were synthesized using the SP6 mMessage mMachine kit (ThermoFisher Scientific) from 2 ug of linearized plasmid following manufacturer's instructions. The DNA template was then removed by the addition of 2 µl of

TURBO DNase and incubation at 37°C for 15 min. Subsequently, synthesized capped mRNAs were purified on the RNeasy kit columns (Qiagen), quantified on a Nanodrop (ThermoFisher Scientifics) and their quality assessed on a 2% agarose gel.

### Injections into zebrafish embryos
300 pg of mRNA or a mixture of morpholinos (Prep1.1, Pbx2 and Pbx4 at 2.7 ng each) mixed with water and 0.1% phenol red dye were injected into 1 to 2 cell stage zebrafish embryos. Following the injection, embryos were raised to the desired time point and used for experimental procedures.

## Assessment of TALE loss of function phenotype
For whole-mount immunostaining, 48hpf embryos were fixed in 4% paraformaldehyde/8% sucrose/1x PBS overnight. Fluorescent staining with the 3A10 primary antibody (1:100; Developmental Studies Hybridoma Bank) and the goat anti-mouse Alexa Fluor 488 secondary antibody (1:200; Molecular Probes A11001) was used to detect Mauthner neurons. For assessment of cartilage formation, 5dpf embryos were fixed in 4% paraformaldehyde/1X PBS overnight, bleached in 30% hydrogen peroxide for 2 hr and stained overnight in 1% HCL/70% ethanol/0.1% alcian blue.

## Identification of in vivo TF binding sites
### ChIP-seq
Groups of 500 zebrafish embryos (total of 10,000 at 3.5hpf and 5000 at 12hpf per biological replicate) were dissociated in 1XPBS by pipetting and fixed for 10 min in 1% formaldehyde. Fixation was stopped by the addition of glycine to a final concentration of 125 mM and cells were pelleted and frozen in liquid nitrogen. Subsequently, cell pellets were processed following a ChIP protocol described previously (Amin et al., 2015). Nuclei were extracted by the addition of 500 µl L1 buffer (50 mM Tris-HCl pH8.0, 2 mM EDTA, 0.1% NP-40, 10% glycerol, 1 mM PMSF) followed by incubation for 5 min on ice and pelleted by centrifugation (3000 rpm, 5 min at 8°C). Nuclei were lysed in 300 µl SDS lysis buffer (50 mM Tris-HCl pH8.0, 10 mM EDTA, 1% SDS) and chromatin sheared into smaller fragments (300 bp on average) by 3 rounds of sonication with a Palmer sonicator (10 s ON – 2 s OFF for a total of 1 min per round, amplitude 40%).

Samples were diluted 10 times in dilution buffer (50 mM Tris-HCl pH8.0, 5 mM EDTA, 200 mM NaCl, 0.5% NP-40, 1 mM PMSF) and pre-cleared by the addition of 50 µl protein-A dynabeads (ThermoFisher Scientific) and incubation for 3 hr at 4°C. After removal of the beads, 10 ul of anti-Prep or pre-bleed antiserum was added (Key Resources Table). Immune complexes were precipitated by the addition of 50 µl of protein-A dynabeads (ThermoFisher Scientific) and incubated for 3 hr at 4°C. Beads were washed five times in wash buffer (20 mM Tris-HCl pH8.0, 2 mM EDTA, 500 mM NaCl, 1% NP-40, 0.1% SDS, 1 mM PMSF), three times in LiCl buffer (20 mM Tris-HCl pH8.0, 2 mM EDTA, 500 mM LiCl, 1% NP-40, 0.1% SDS, 1 mM PMSF) and three times in TE buffer (10 mM Tris-HCl pH8.0, 1 mM EDTA, 1 mM PMSF).

Chromatin fragments were eluted by the addition of 50 µl of freshly made elution buffer (10 mM Tris-HCl pH8.0, 1 mM EDTA, 2% SDS) and incubation at 25°C for 15 min followed by an incubation at 65°C for another 15 min. Then, DNA fragments were reverse cross-linked by adding 2.5 µl of 5M NaCl and incubating at 65°C O/N. Finally, DNA fragments were recovered in 10 µl nuclease free water using a PCR purification mini-elute kit (Qiagen).

ChIP DNA fragments and their corresponding input were quantified on a Qubit with the dsDNA HS assay kit (ThermoFisher Scientific). 10 ng of DNA was used for library preparation using the Tru-seq ChIP Sample Preparation Guide (Illumina Inc). For samples containing less than 10 ng of DNA the entire eluted DNA was used. Briefly, sample DNA was blunt-ended and phosphorylated, and a single 'A' nucleotide added to the 3' ends of the fragments in preparation for ligation to an adapter with a single-base 'T' overhang. Omitting the size selection step, the ligation products were then PCR-amplified to enrich for fragments with adapters on both ends. Libraries were sequenced on an Illumina HiSeq2500 Sequencer.

### ChIP-qPCR
The ChIP protocol for ChIP-qPCR is the same as described in the ChIP-seq section above except that a total of 1000 wild-type or injected embryos were collected for NF-YB and Pbx4 ChIPs and 200

embryos for Histone H3 and H3K27ac ChIPs. The following antibodies were used: 10 µl of anti-Prep1.1 and anti-Pbx4 in house sera and their corresponding pre-bleed control sera; 8 µg of anti-NF-YB rabbit polyclonal antibody and control rabbit polyclonal IgG. The relative quantification of select genomic regions was determined by qPCR using specific primers pairs (see *Supplementary file 5*) and 2 µl of ChIP DNA eluate.

## Quantification of gene expression

### Total RNA extraction from zebrafish embryos

Total RNA from 50 to 100 6hpf or 12hpf zebrafish whole embryos was extracted with the RNeasy kit (Qiagen) following manufacturer's instructions. Total RNA was then used in RNA-seq and RT-qPCR reactions.

### RNA-seq

Total RNA quantification and quality assessment was performed on a Bioanalyzer (Agilent) and only total RNAs with a RNA Integrity Number above nine were further considered. Then, 3 ug of total RNA was used to construct RNA-seq libraries with the Illumina Truseq stranded mRNA library kit after PolyA + RNA enrichment. The quality and size of the fragments was determined on a Bioanalyzer (Agilent) and single-end 100 bp reads were generated on a Hi-Seq sequencer at the molecular biology core of the University of Massachusetts Medical School.

### RT-qPCR

500 ng to 1 µg of total RNA was reverse transcribed using the high capacity cDNA kit (ThermoFisher Scientific). The relative quantity of select mRNAs was determined by qPCR: each 25 ul total PCR reaction contained 2 µl of cDNA diluted 10-fold, 0.2 mM of each specific primer (see *Supplementary file 5*) and qPCR master mix (Biotool) to a 1X final concentration. The reactions were loaded onto a 7300 real-time PCR system (Applied Biosystems).

## Generation of expression vectors

Myc-Prep1.1 (NM_131891.3), HA-Pbx4 (NM_131447.1) encoding plasmids were described previously (*Choe et al., 2009*, *2002*). Flag-NF-YA and Flag-NF-YB plasmids were generated by PCR amplification of the zebrafish NF-YA (NM_001082795.1) and NF-YB (NM_001013322.2) coding sequences from 24hpf zebrafish cDNA using specific primers bearing EcoRI/XhoI and XbaI/SnabI restriction sites respectively. The amplified sequences were then introduced into a PCS2 + plasmid backbone. Subsequently, a Flag tag sequence was PCR amplified from a p3xFLAG-CMV−7.1 vector using specific primers bearing EcoRI (for NF-YA) or StuI/XbaI (for NF-YB) and cloned 5' to the NF-YA or B coding sequences. The NF-YDN plasmid was constructed as previously described (*Mantovani et al., 1994*). Briefly, three point mutations ($R^{279}A$, $G^{280}A$, $D^{281}A$) located in the conserved NF-YA DNA binding domain, preventing NF-YA DNA binding but not interactions with the other members of the NF-Y complex, were introduced using the Q5 site directed mutagenesis kit (New England Biolabs) and primers bearing the mutations. Plasmids for luciferase reporter assays were generated by amplifying ~500 bp genomic fragments containing the Prep binding sites associated with the *tle3a*, *pax5*, *prdm14*, *tcf3a*, *her6*, *dachb* and *fgf8* loci (using the primers listed in *Supplementary file 5*) and cloning into the XhoI sites of the pGL3-Promoter vector (Promega E1761)

All the plasmids were validated by Sanger sequencing, amplified in DH5α bacterial cells and extracted using the PureLink HiPure Plasmid Midiprep Kit (ThermoFisher Scientific). All primer sequences can be found in *Supplementary file 5*.

## Luciferase assays and assessment of protein-protein interactions

### Transfection

$3 \times 10^6$ HEK-293T cells were seeded on 10 cm dishes and allowed to grow overnight in antibiotic-free growth medium (DMEM (Gibco) supplemented with 10% FBS (Hyclone)). HEK293T cells were obtained from ATCC (ATCC CRL-3216). These cells were not independently authenticated and were not tested for mycoplasma. The next day, the cells were incubated for 5 hr in Opti-MEM (Gibco) medium containing a mixture of plasmid DNA and Lipofectamine 2000 (Invitrogen) following

manufacturer's instructions. Subsequently, the cells were incubated overnight in fresh antibiotic-free growth medium.

## Immunoprecipitation of TALE-NF-Y protein complexes

Transfected cells were lysed in 4 mL of ice cold Co-IP Buffer (50 mM Tris-HCl pH 7.5, 150 mM NaCl, 0.2 mM EDTA, 1 mM DTT, 0.5% Triton X100, 1X Complete Protease Inhibitor (Roche)) and incubated on ice for 30 min. Cell lysates were centrifuged at 2,000 g for 10 min at 4°C to remove cell debris and pre-cleared by incubation at 4°C after the addition of 50 µL of Protein A/G Agarose Beads blocked in 1% BSA for 1 hr (Roche). To immunoprecipitate the target protein, 8 µg of the appropriate antibody (see Key Resources Table) was added to each sample before incubation at 4°C overnight. The next morning 40 µL of Protein A/G Agarose beads blocked with 1% BSA was added and each sample incubated for 4 hr at 4°C. Non-specific binding was eliminated by five washes in 1 mL of Co-IP Buffer. Finally, the immune-complexes were eluted in 80 µL of 1X Laëmmli Buffer (Biorad) containing 2.5% beta-mercaptoethanol and agitated for five minutes at 95°C.

## Western blot

20 µL of each IP sample or 13 µL of each Input sample were loaded onto a 4–20% gradient polyacrylamide gel (Bio-Rad) and the proteins separated at 200V until the dye front reached the end of the gel. The separated proteins were then transferred onto a methanol-activated PVDF membrane at 100V for one hour. After incubation for one hour in blocking buffer (5% non-fat dehydrated milk in Tris Buffered Saline with Tween (TBST; 50 mM Tris-HCl pH 7.5, 150 mM NaCl, 0.1% Tween 20)) the membranes were probed with specific antibodies (see Key Resources Table) diluted in TBS-Tween plus 5% BSA and incubated overnight at 4°C. The next day after four washes of 10 min in TBS-Tween the membrane were probed with the appropriate secondary antibody diluted in TBS-Tween plus 5% BSA and incubated at 4°C for two hours. After four washes of ten minutes in TBS-Tween the ECL reaction was performed and chemiluminescence detected with a LAS3000 (Fuji) machine.

## Luciferase reporter assay

For the reporter assays, 100 or 400 ng of each luciferase reporter plasmid was co-transfected (see above for transfection protocol) with 200 ng of each TF (Meis, Pbx, NF-YA and NF-YB) or with 800 ng of control plasmid, as well as together with 50 ng of a plasmid expressing *renilla* luciferase. Luciferase was quantified using the DualGlo Luciferase system (Promega E2920) in a Perkin Elmer Envision 2104 Multiplate reader and firefly luciferase levels were normalized to *renilla* levels. Each assay was performed in triplicate and is presented as mean fold induction ± SD over transfection with empty vector. A vector containing the SV40 enhancer (pGL3-Control vector; Promega E1741) was used as positive control.

## Quantification and statistical analysis

Analysis of expression and ChIP data was done as outlined below using standard bioinformatics packages. Default statistics tools included in each package were used (except as indicated) and the exact parameters for each type of analysis are listed below.

## Processing of RNA-seq data

Fastq files containing strand specific trimmed and filtered reads were processed using the University of Massachusetts Medical School Dolphin web interface (see Key Resources Table). Reads were quality checked with FastQC aligned to the DanRer10 zebrafish transcriptome and normalized gene expression TPM (Transcripts Per Million) values calculated using RSEM_v1.2.28 with parameters -p4 –bowtie-e 70 –bowtie-chunkmbs 100 (*Li and Dewey, 2011*). Identification of differentially expressed genes (DEGs) was performed with DeSeq2 (*Anders and Huber, 2010*) on three independent biological replicates for each control or TALE KD conditions except for RNA-seq data of TALE KD vs Control embryos at 12hpf. In this latter experiment one outlier replicate was excluded from the analysis. DeSeq2 identified DEG with $p$-adj ≤0.05 (Benjamini and Hochberg FDR) and to compensate for the loss of one biological replicate only DEGs with $p$-adj ≤0.01 were used in all subsequent analyses.

## Processing of ChIP-seq data

Fastq files for ChIP-seq analysis contained 101 bp paired-end sequence for Prep 3.5hpf and 12hpf, two biological replicates each, and matched input-DNA controls. After an assessment of the raw sequence quality using FastQC (*Babraham Institute. n.d, 2016*) and Fastq-screen (*Babraham Institute. n.d, 2016*) the sequence reads were filtered to remove any remaining adapter sequence or poor quality 3' end sequence using Trimmomatic version 0.32 (*Bolger et al., 2014*). Default parameters for ILLUMINACLIP and SLIDINGWINDOW were used. MINLENGTH was set to 50 bp, except for Prep 3.5hpf replicate 2 with which 36 bp was used. The reads were then mapped to the GRCz10 (danRer10/September 2014) release of the entire zebrafish genome from the UCSC browser (*Tyner et al., 2017*) using Bowtie2 version 2.2.3 (*Langmead and Salzberg, 2012*). The output SAM file was further filtered to remove reads with poor mapping quality and discordant mapped read pairs, using SAMtools view version 0.1.19 (*Li et al., 2009*) (with flags used -f 2 -q30). Peak calling was performed using MACS2 version 2.1.0.20140616 (*Zhang et al., 2008*), excluding reads that mapped to the mitochondrial genome and unassembled contigs in the assembly. Default parameters were used, except that the effective genome size was set to 1.03e9 (this equates to 75% of the total genome sequence, excluding 'N' bases. The *q*-value threshold was set to 0.05. Candidate binding regions were then filtered to retain those with a fold enrichment of $\geq$10. Upon applying these criteria, we noticed that one biological replicate for each ChIP-seq experiment (3.5hpf and 12hpf) underperformed, but more than 95% of the peaks were identified also in the second biological replicate (see '*Quantification of ChIP peak overlap*' below and *Supplementary file 1*). Therefore, the best biological replicate for each experimental condition was considered for downstream analysis.

## Analysis of qPCR results

### Gene expression analysis

Gene expression was determined and normalized to *gapdh* expression using the following formula $(0.5^{\text{gene of interest Ct value}}/0.5^{\text{gapdh Ct value}})$. The mean value and standard error of the mean (SEM) for three independent biological replicates of control and experimental conditions were calculated using Excel. Statistical significance of mean variations between two conditions was calculated using an unpaired t-test in Excel. Two conditions are considered significantly different if p-value$\leq$0.05.

### ChIP DNA enrichment analysis

DNA enrichment was determined and normalized to input values using the following formula $(0.5^{\text{IP Ct value}}/0.5^{\text{Input Ct value}})$. Then the mean value and standard error of the mean (SEM) for three independent biological replicates of control and experimental conditions were calculated using Excel. When necessary the results were expressed as a fold change of specific ChIP signal over control IgG ChIP signal. Statistical significance of mean variations between two conditions was calculated using an unpaired t-test in Excel. Two conditions are considered significantly different if p-value$\leq$0.05.

## Analysis of GO term enrichment

GREAT (version 3.0.0 [*McLean et al., 2010*; *Hiller et al., 2013*]) allowed for the analysis of GO term enrichment using Prep binding site coordinates as Input. The analysis was performed using the single nearest gene within 5 or 30 kb association rule since most Prep sites are found within 30 kb of a TSS. GO terms were ranked by Binomial False Discovery Rate (FDR) values. The results are presented as -log2 transformed FDR values and only GO terms with FDR $\leq$ 0.05 (-log2(FDR) $\leq$ 4.32) were considered significant.

DAVID (version 6.8 [*Huang et al., 2009b*, *2009a*]) was used to identify enriched GO terms associated with genes identified in the RNA-seq analysis and/or found to be near Prep binding sites. The Benjamini multiple testing False Discovery Rate (FDR) was use to rank the identified GO terms. The results are presented as -log2 transformed FDR values and only GO terms with FDR $\leq$ 0.05 (-log2 (FDR) $\leq$ 4.32) were considered significant.

## Analysis of TF peak features

All TF binding site coordinates used in the following analysis were defined as 200 bp coordinates centered on the ChIP peak summit. Unless otherwise specified, only peaks with an FE $\geq$ 10 were considered.

## Prep binding sites distribution relative to TSSs

The distribution of Prep binding sites relative to TSSs was calculated using the windowbed tool from the bedtools suite (*Quinlan and Hall, 2010*) in the Galaxy toolshed (*Goecks et al., 2010*) searching for the number of Prep binding sites found within 5 or 30 kb (from their center) of any Ensembl zebrafish (Zv9) or mouse (Mm9) TSSs.

## Identification of prep peak associated genes

A gene was considered associated with a Prep binding site if any of its Ensembl (Zv9) TSS was found within 5 or 30 kb from a Prep peak. Prep-associated genes were defined using the windowbed tool from the Bedtools suite in Galaxy searching for Ensembl TSS (for instance those of differentially expressed genes in TALE KD embryos or first-wave wave genes) found within 5 or 30 kb of the center of any Prep binding site. Statistical significance of Prep binding association with genes of interest (first wave genes and TALE KD differentially expressed genes) over a random population of genes was determined with a Pearson correlation test with a statistical significance $\leq$0.05.

## Quantification of ChIP peak overlap

The overlap between two populations of ChIP peaks was analyzed using the intersect tool from the Galaxy toolshed. Two Prep peaks (in different ChIP biological replicates or in ChIP-seq results from 3.5hpf vs. 12hpf) were considered to overlap if their summits were within 50 bp (See also Processing of ChIP-seq data above). Prep and NF-YA peaks in mESCs were considered to overlap if their summits were within 500 bp.

## Identification of the Prep$_{12hpf-only}$ peak population

Prep$_{12hpf-only}$ ChIP-seq peaks were identified by subtracting Prep$_{12hpf}$ peaks overlapping with all Prep1$_{3.5hpf}$ peaks identified by MACS2 without applying any enrichment cut-off. This strategy allowed for stringent identification of 11468 Prep$_{12hpf-only}$ binding sites not occurring at 3.5hpf that were used for subsequent analysis.

## TF binding motif analysis

MEME and DREME (MEME-suite version 4.11.1 [*Machanick and Bailey, 2011*; *Bailey et al., 2009*]) were used to identify significantly enriched de novo binding motifs. DREME ran in a default mode, MEME was set to search for a maximum of six 4 to 12 nucleotide long motifs. Motif distribution relative to ChIP-seq peak summit was defined by CENTRIMO using default parameters. AME (MEME-suite version 4.11.1 [*Machanick and Bailey, 2011*; *Bailey et al., 2009*]) was used to calculate the relative enrichment between two datasets using default parameters (Ranksum test, p-value$\leq$0.05). In the case of a relative enrichment against a control set of sequence, the « shuffled input sequences » mode was selected. The occurrence of TF binding motifs in *Figure 3D* and *Figure 7—figure supplement 1D*) was calculated using a custom Python script (moth.py, *Source code 1*) with the input files provided in *Figure 3—source data 1*. To do so, regular expression matches were identified on both strands of the input sequences, and the number of sequences containing at least one occurrence of a motif was calculated. HEXA motifs were identified in sequences that did not contain any DECA motif.

## Sequence conservation analysis

Average conservation score around Prep1 binding sites was computed in the Deeptool suite using Prep1-bound sequences and the UCSC vertebrate PhastCons eight way (Zebrafish, Medaka, Stickleback, Tetraodon, Fugu, X. tropicalis, Mouse, Human) wig file as regions of interest and score input files respectively. For *Figure 4—figure supplement 1A*, a set of 11000 random chromosomal coordinates was generated from the zv11 zebrafish genome assembly using the randCoord.py custom python script (*Source code 2*).

## Analysis of chromatin features

Chromatin heatmaps and mean score profiles of Prep binding sites in fish embryos and mESCs were generated with the Deeptools (version 2.0 [*Ramírez et al., 2014*]) suite of tools in the Galaxy

toolshed. BED files containing Prep biding site coordinates and wiggle files of previously published datasets (Key Resources Table) downloaded from GEO or ENCODE were used as inputs. First, signal matrices at Prep bound regions were made using the compute matrix tool in reference-point mode with the following parameters: distance upstream and downstream of the start site of the regions defined in the BED file: 1000 or 2,000 bp, bin size: 25 bp. When necessary, the regions were ranked based on mean signal values. Second, score matrices were used to generate heatmaps and mean score profiles with the plot heatmaps and plot profile tools respectively. We note that the public ChIP-seq and ATAC-seq datasets are from slightly different timepoints (4.5hpf and 4hpf, respectively) than our Prep ChIP-seq dataset (3.5hpf). Since each dataset requires hundreds to thousands of embryos (that cannot be individually staged) and zebrafish development is slightly asynchronous, it is likely that collecting embryos at these three timepoints will result in considerable overlap of the actual stages analyzed.

## Acknowledgements

We are grateful to the Genomic Technologies and Bioinformatics Core Facilities at the University of Manchester, UK and to Alper Kucukural at the University of Massachusetts Bioinformatics Core for assistance.

## Additional information

### Funding

| Funder | Grant reference number | Author |
| --- | --- | --- |
| National Institute of Neurological Disorders and Stroke | NS38183 | Charles G Sagerström |
| Biotechnology and Biological Sciences Research Council | BB/N00907X/1 | Nicoletta Bobola |

The funders had no role in study design, data collection and interpretation, or the decision to submit the work for publication.

### Author contributions

Franck Ladam, Conceptualization, Formal analysis, Investigation, Visualization, Writing—review and editing; William Stanney, Ozge Yildiz, Investigation, Writing—review and editing; Ian J Donaldson, Formal analysis, Writing—review and editing; Nicoletta Bobola, Conceptualization, Supervision, Funding acquisition, Writing—review and editing; Charles G Sagerström, Conceptualization, Supervision, Funding acquisition, Writing—original draft, Project administration, Writing—review and editing

### Author ORCIDs

Nicoletta Bobola (ID) http://orcid.org/0000-0002-7103-4932
Charles G Sagerström (ID) http://orcid.org/0000-0002-1509-5810

### Ethics

Animal experimentation: This study was submitted to and approved by the University of Massachusetts Medical School Institutional Animal Care and Use Committee (protocol A-1565) and the University of Massachusetts Medical School Institutional Review Board (protocol I-149).

### Decision letter and Author response

Decision letter https://doi.org/10.7554/eLife.36144.065
Author response https://doi.org/10.7554/eLife.36144.066

## Additional files

**Supplementary files**

• Source code 1. moth.py (Motif Occurrence for TALE and Hox factors) file. This module identifies transcription factor motifs (for TALE, Hox and NF-Y) in input sequences. The input sequences are provided as *Figure 3—source data 1*.
DOI: https://doi.org/10.7554/eLife.36144.016

• Source code 2. randCoord.py file. This module generates a set of random chromosome intervals of specified length from an input genome sequence. The input sequence used was the full zv11 zebrafish genome sequence.
DOI: https://doi.org/10.7554/eLife.36144.017

• Supplementary file 1. ChIP-seq statistics. Related to *Figure 2*. Prep ChIP-seq experiments were performed on 3.5hpf and 12hpf zebrafish embryos. Biological replicates (n = 2) for each condition show a high degree of reproducibility as shown by the percent of peaks found in both replicates (replicate overlap). Peaks with Fold Enrichment (FE) $\geq$ 10 were used for subsequent analysis.
DOI: https://doi.org/10.7554/eLife.36144.018

• Supplementary file 2. GO terms associated with Prep MPAD populations. Related to *Figure 4D*. GREAT analysis (association rule = single nearest gene within 30 kb) was used to identify GO-terms for each MPAD population. Significant GO terms (FDR $\leq$ 0.05; -log2FDR $\leq$ 4.32) associated with any MPAD population were then merged using the 'join two files' tool in Galaxy to produce a list of 231 GO-terms associated with a –log2 (FDR) value for each MPAD population. Finally, GO terms and their corresponding –log2 (FDR) values were grouped (1-8) based on their association with the various MPAD populations. Note that Class 4 MPADs are enriched near genes involved in embryonic development, including processes related to the TALE loss-of-function phenotype (nervous system, eye and heart development; Group 5). N.S. = not significant.
DOI: https://doi.org/10.7554/eLife.36144.019

• Supplementary file 3. GO-term enrichment analysis. Related to *Figures 1*, *2* and *6*, *Figure 4—figure supplement 2* and *Figure 5—figure supplement 1*. *Figures 1D*, *2E* and *6C*, *Figure 4—figure supplement 2C* and *Figure 5—figure supplement 1D* show only a representative set of GO-terms for each analysis. This table lists all GO-terms identified by each GREAT or DAVID analysis in this study. Details of each analysis can be found in the first tab of the table.
DOI: https://doi.org/10.7554/eLife.36144.020

• Supplementary file 4. Information on TALE GRN genes associated with Class 4 MPADs. Related to *Figure 6*.
DOI: https://doi.org/10.7554/eLife.36144.021

• Supplementary file 5. Primer sequences used in this study.
DOI: https://doi.org/10.7554/eLife.36144.022

• Transparent reporting form
DOI: https://doi.org/10.7554/eLife.36144.023

### Data availability

RNA-seq data has been deposited in GEO under accession code GSE102662 ChIP-seq data has been deposited in ArrayExpress under accession code E-MTAB-5967

The following datasets were generated:

| Author(s) | Year | Dataset title | Dataset URL | Database, license, and accessibility information |
|---|---|---|---|---|
| Ladam F, Sager-strom CG | 2018 | Zebrafish TALE KD RNA-seq | https://www.ncbi.nlm.nih.gov/geo/query/acc.cgi?acc=GSE102662 | Publicly available at the NCBI Gene Expression Omnibus (accession no: GSE102662) |
| Ladam F, Stanney W, Donaldson IJ, Bobola N, Sager-strom CG | 2017 | ChIP-seq for Prep on whole zebrafish embryos at 3.5 and 12hp | http://www.ebi.ac.uk/arrayexpress/experiments/E-MTAB-5967 | Publicly available at the Electron Microscopy Data Bank (accession no: |

E-MTAB-5967)

The following previously published datasets were used:

| Author(s) | Year | Dataset title | Dataset URL | Database, license, and accessibility information |
|---|---|---|---|---|
| Hans-Jörg Warnatz | 2015 | Prep1 (ChIP-Seq) | https://www.ncbi.nlm.nih.gov/geo/query/acc.cgi?acc=GSM1545025 | Publicly available at the NCBI Gene Expression Omnibus (accession no: GSM1545025) |
| Ozren Bogdanovic | 2012 | H3K4me1_dome, danRer7 | https://www.ncbi.nlm.nih.gov/geo/query/acc.cgi?acc=GSM915193 | Publicly available at the NCBI Gene Expression Omnibus (accession no: GSM915193) |
| Ozren Bogdanovic | 2012 | H3K4me3_dome, danRer7 | https://www.ncbi.nlm.nih.gov/geo/query/acc.cgi?acc=GSM915189 | Publicly available at the NCBI Gene Expression Omnibus (accession no: GSM915189) |
| Ozren Bogdanovic | 2012 | H3K27ac_dome, danRer7 | https://www.ncbi.nlm.nih.gov/geo/query/acc.cgi?acc=GSM915197 | Publicly available at the NCBI Gene Expression Omnibus (accession no: GSM915197) |
| Ozren Bogdanovic | 2012 | H3K27ac_80%epi, danRer7 | https://www.ncbi.nlm.nih.gov/geo/query/acc.cgi?acc=GSM915198 | Publicly available at the NCBI Gene Expression Omnibus (accession no: GSM915198) |
| Ozren Bogdanovic | 2012 | H3K27ac_24hpf, danRer7 | https://www.ncbi.nlm.nih.gov/geo/query/acc.cgi?acc=GSM915199 | Publicly available at the NCBI Gene Expression Omnibus (accession no: GSM915199) |
| Yong Zhang | 2013 | H3K27me3 ChIP-seq dome | https://www.ncbi.nlm.nih.gov/geo/query/acc.cgi?acc=GSM1081557 | Publicly available at the NCBI Gene Expression Omnibus (accession no: GSM1081557) |
| Yong Zhang | 2013 | nucleosome dome rep 1 | https://www.ncbi.nlm.nih.gov/geo/query/acc.cgi?acc=GSM1081554 | Publicly available at the NCBI Gene Expression Omnibus (accession no: GSM1081554) |
| Yong Zhang | 2013 | Pol II ChIP-seq dome 8WG16 | https://www.ncbi.nlm.nih.gov/geo/query/acc.cgi?acc=GSM1081560 | Publicly available at the NCBI Gene Expression Omnibus (accession no: GSM1081560) |
| Hyung Joo Lee | 2015 | MeDIP_4.5hpf | https://www.ncbi.nlm.nih.gov/geo/query/acc.cgi?acc=GSM1274386 | Publicly available at the NCBI Gene Expression Omnibus (accession no: GSM1274386) |
| Raja Jothi | 2014 | ChIP-Seq NF-YA | https://www.ncbi.nlm.nih.gov/geo/query/acc.cgi?acc=GSM1370111 | Publicly available at the NCBI Gene Expression Omnibus (accession no: GSM1370111) |
| ENCODE DCC | 2012 | LICR_ChipSeq_ES-E14_H3K4me1_E0 | https://www.ncbi.nlm.nih.gov/geo/query/acc.cgi?acc=GSM1000121 | Publicly available at the NCBI Gene Expression Omnibus (accession no: |

| | | | | GSM1000121) |
|---|---|---|---|---|
| ENCODE DCC | 2012 | LICR_ChipSeq_ES-E14_H3K4me3_E0 | https://www.ncbi.nlm.nih.gov/geo/query/acc.cgi?acc=GSM1000124 | Publicly available at the NCBI Gene Expression Omnibus (accession no: GSM1000124) |
| ENCODE DCC | 2012 | LICR_ChipSeq_ES-E14_H3K27ac_E0 | https://www.ncbi.nlm.nih.gov/geo/query/acc.cgi?acc=GSM1000126 | Publicly available at the NCBI Gene Expression Omnibus (accession no: GSM1000126) |
| ENCODE DCC | 2012 | LICR_ChipSeq_ES-Bruce4_H3K27me3_E | https://www.ncbi.nlm.nih.gov/geo/query/acc.cgi?acc=GSM1000089 | Publicly available at the NCBI Gene Expression Omnibus (accession no: GSM1000089) |
| ENCODE DCC | 2012 | UW_DnaseSeq_ES-E14_E0_129/Ola | https://www.ncbi.nlm.nih.gov/geo/query/acc.cgi?acc=GSM1014154 | Publicly available at the NCBI Gene Expression Omnibus (accession no: GSM1014154) |
| Chieh-Chun Chen | 2014 | E14 MeDIP-seq | https://www.ncbi.nlm.nih.gov/geo/query/acc.cgi?acc=GSM859494 | Publicly available at the NCBI Gene Expression Omnibus (accession no: GSM859494) |
| Hans-Jörg Warnatz | 2015 | Input_DNA (ChIP-Seq control) | https://www.ncbi.nlm.nih.gov/geo/query/acc.cgi?acc=GSM1545026 | Publicly available at the NCBI Gene Expression Omnibus (accession no: GSM1545026) |

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
