## [Decision Letter]

Thank you for submitting your article "TALE factors use two distinct functional modes to control an essential zebrafish gene expression program" for consideration by *eLife*. Your article has been reviewed by three peer reviewers, and the evaluation has been overseen by Marianne Bronner as the Senior/Reviewing Editor. The following individuals involved in review of your submission have agreed to reveal their identity: Hugo Parker (Reviewer #1); Licia Selleri (Reviewer #2); Miguel Torres (Reviewer #3).

The reviewers have discussed the reviews with one another and the Reviewing Editor has drafted this decision to help you prepare a revised submission. The essential revisions are described below but we also include the full reviews of the three reviewers for further details.

Summary:

In this manuscript, Ladam et al. investigate how TALE factors contribute to early zebrafish development by characterizing Prep1.1 DNA-binding profiles at blastula and segmentation stages. They correlate these binding profiles with chromatin marks at these loci and with gene expression data from wild-type vs. TALE knock-down embryos. By focusing on two developmental timepoints, they make the interesting finding that the DNA-binding preference of Prep1.1 appears to expand during development. Strikingly, early binding to 'DECA' motifs is a feature at the blastula stage and additional binding to 'HEXA' motifs with adjacent Pbx-Hox motifs is seen by the segmentation stage. Exploring this further, they identify NF-Y as a potential binding partner of Pbx and Prep in blastula embryos. This leads to a model in which TALE factors utilise different modes of DNA-binding at different times in development, depending on the availability of co-factors such as NF-Y and Hox factors.

Essential revisions:

1) The authors must demonstrate that their model is correct by testing several of the identified elements by reporter assays in zebrafish embryos (or at the very least in cell lines in vitro), coupled with mutation of predicted TALE and NF-Y binding sites to address the importance of these sites for enhancer function.

2) The TALE knock-down phenotype needs to be better characterized with appropriate validation of the specificity of the morpholino.

3) The authors need to explain the temporal discrepancies of all their assays.

4) The authors should provide clarifications/explanations regarding the statement that TALE GRN genes are significantly associated with Class 4 and Class 3, but not Class 1 or 2, MPADs (Figure 6A, B).

5) The authors should include a final diagram depicting the model of Prep1.1/TALE DNA-binding dynamics across developmental time and how this relates to the activation of the components of the TALE-GRN, changes in chromatin state and interactions with co-factors.

6) Absolute statements should be avoided (e.g. "TALE factors control gene expression by regulating a chromatin transition… at a core set of genes encoding TFs that direct anterior development.")

*Reviewer #1:*

This work addresses an important question because Prep and Pbx proteins are crucial factors for vertebrate development and are present throughout embryogenesis but most of the focus up to now has been on their roles during segmentation stages, with their early roles and mode/s of DNA-binding receiving relatively little attention. The finding that these factors exhibit an expansion of their DNA-binding repertoire between blastula and segmentation stages is novel and interesting, representing a significant advance in our understanding of the dynamic roles of these factors during early development. However, I have a few points that I think should be addressed, that would considerably strengthen the conclusions and the manuscript.

i) The TALE knock-down phenotype needs to be described/characterised in more detail, to provide a more comprehensive view of the developmental context and to validate the specificity of the morpholino cocktail. For instance, hindbrain segmentation/neuroanatomy and craniofacial morphology should be characterised in the triple morphants to provide more detailed evidence that the knockdown is as expected given the previously characterised MO phenotypes. I also suggest moving the justification for using MO's that is in the Figure 1—figure supplement 1 legend to the Materials and methods section or the main text, where it will be more prominent.

ii) An assumption made is that Prep1.1-bound sites, or at least a sub-set of them, represent enhancer elements. The authors must demonstrate that this is true by testing a few such elements by reporter assay in zebrafish embryos, coupled with mutation of predicted TALE and NF-Y sites to address the importance of these sites for enhancer function. This can be done relatively quickly by transient transgenesis, is frequently used for mechanistic dissection of cis-regulatory elements, and will provide crucial evidence for the functionality of these putative enhancers.

iii) The authors use data from mouse ESCs to infer evolutionary conservation of the TALE GRN and of TALE-NF-Y co-localised binding, which expands the scope beyond zebrafish. A complementary approach is to address how many Prep1.1-bound peaks overlap with fish-mammal conserved non-coding elements that have been described in the literature. It is also worth checking if any are homologous to elements in the VISTA enhancer browser and have been experimentally validated in transgenic mice. This is straightforward to do and could potentially add weight to the argument that these interactions are evolutionarily conserved.

iv) This manuscript would really benefit from a final diagram depicting the model of Prep1.1/TALE DNA-binding dynamics across developmental time and how this relates to the activation of the components of the TALE-GRN, changes in chromatin state and interactions with co-factors.

*Reviewer #2:*

The paper by Franck Ladam et al. presents original studies on unexplored mechanisms that underlie very early roles of TALE transcription factors (TFs) in blastula/gastrula versus later roles during segmentation stages of zebrafish development.

This interesting study makes a substantial leap forward by identifying binding of TALE factors at genomic Pbx:Prep (DECA) sites during early zebrafish developmental stages, at 3.5 hpf. The authors also report that binding motifs for the maternal NF-Y TF are enriched near DECA sites and that NF-Y can form complexes with TALE proteins. Interestingly, the authors demonstrate that in the later post-gastrula embryo, at segmentation stages, GRN-associated TALE occupancy expands to include HEXA motifs with adjacent PBX:HOX sites. Therefore, the authors convincingly demonstrate that TALE factors control a key GRN, but utilize distinct DNA motifs and protein partners at different developmental stages, which is a novel and as yet unexplored mechanism underlying differential and temporally-restricted functions of TALE TFs in vertebrate embryogenesis.

The manuscript comprises an arsenal of high-quality results that are illustrated in complex and impeccable figures. The identification of distinct sequence-based mechanisms that underlie TALE binding to DNA thus directing different developmental functions at successive stages of embryogenesis is per se a fundamental finding. This study will be of high importance and interest for TALE biology, which sorely lacks mechanistic research conducted in vivo in animal models. The original findings reported in this paper can also open newpaths of investigation on roles of TALE factors in the earliest stages of embryogenesis in other organisms, including mammals. In addition, the distinct strategies that are being employed by TALE factors to execute developmental processes at different stages of embryogenesis might be similarly adopted by other TFs with widespread expression patterns, thus opening additional avenues for broader investigations.

Concerns that should be addressed:

1) Results – General Consideration:

The authors conduct multiple genome-wide experiments and/or mine available genome-wide datasets, including RNA-Seq, ChIP-seq for Prep1, ATAC Seq, ChIP-seq for chromatin marks. The time-point analyzed are not fully consistent across these experimental approaches, e.g. RNA-Seq for controls and TALE KD zebrafish embryos is performed at 6hpf (early gastrula) and at 12hpf (segmentation stage); ChIP-seq for Prep1.1 is performed at 3.5hpf (blastula) and at 12hpf (segmentation stage); ATAC-Seq at 4hpf (blastula; available datasets); and ChIP-seq for chromatin marks at 4.5hpf (blastula; available datasets). While this could be somewhat concerning, the findings and the overall message emerging from the study are strong and do not appear to be weakened by the slight temporal discrepancies. Indeed: a) out of the 13,300 Prep peaks at 3.5hpf, ~60% co-localize with a Prep peak at 12hpf, suggesting that a large fraction of binding sites remains occupied throughout embryogenesis; b) an additional ~16,500 peaks detectable at 12hpf do not co-localize with Prep 3.5hpf peaks, demonstrating that additional binding sites become occupied at later developmental stages; c) Prep binding is dynamically and continuously associated with the TALE GRN during zebrafish embryogenesis; d) TALE factors utilize distinct binding motifs at very early versus late stages of embryogenesis; e) TALE-occupied sites are associated with specific chromatin states at blastula stages; f) developmental control genes are enriched near Modified Prep Associated Domains (MPADs) displaying repressive histone modifications; g) Class 4 MPADs transition to an active chromatin state during later stages of embryogenesis; h) TALE and NF-Y factors have joint roles at very early developmental stages and can form complexes.

This reviewer is not particularly concerned about the minor temporal discrepancies present among the various genome-wide assays conducted in this study, given the strong message emerging from all of the reported high-throughput experiments. However, it might be useful to underscore throughout the text and in the Discussion that TALE factors adopt distinct mechanistic strategies in zebrafish blastula and early gastrula versus segmentation stages; in other words to simply cluster together the functions of TALE TFs in blastula and early gastrula within one single group (comprising 3.5, 4, 4.5, and 6 hpf). To this end, it would help to slightly modify the cartoon illustrating the subsequent zebrafish developmental stages in Figure 1A. The authors could group [blastula stages and early gastrula stages] within one single bracket or inside one single box andthe [segmentation stages] inside another bracket or box. Accordingly, this clustering could be clarified in the figure legend.

2) Results, subsection “TALE factors control the chromatin state at Class 4 MPADs associated with the anterior GRN”: The authors state that TALE GRN genes are significantly associated with Class 4 and Class 3, but not Class 1 or 2, MPADs (Figure 6A, B). However, RNA-seq (Figure 5C) shows that genes associated with Class 3 MPADs (and also Class 1 and 2 MPADs) are expressed at similar levels at 12hpf and 6hpf (Figure 5D). In contrast, Class 4 MPADs display higher levels of H3K27ac at 9hpf than at 4.5hpf (Figure 5A, B) and their associated genes show the greatest increase in expression between 6hpf and 12hpf. In addition, only class 4 MPADs showed a strong switch to an active chromatin state from 4.5hpf to 9hpf during zebrafish embryogenesis (Figure 5A, B), while class 3 MPADs did not exhibit any significant switch (Figure 5A, B). Collectively, these results are somewhat difficult to reconcile. The authors should qualify these findings and try to explain these differences. Are other factors necessary for activation of Class 3 MPADs? Or do the acetylation changes appear at a later time-point for Class 3 MPADs? Can other scenarios be envisaged? It would be helpful to add these considerations at least to the Discussion section of the paper.

3) Results: The authors state: "These findings indicate that TALE factors control gene expression by regulating a chromatin transition – from repressive chromatin in blastula stage embryos to active chromatin in segmentation stage embryos – at a core set of genes encoding TFs that direct anterior development."

This statement is very strong and absolute, whereas it does not hold across the entire animal kingdom. In fact, in the mouse TALE factors substantially affect "posterior" development, as shown by the presence of severe posterior developmental defects in various mouse models with LOF for different TALE TFs. For example, in Pbx LOF mouse embryos it has been reported that the posterior axial skeleton, the hindlimb, and the urogenital system are severely compromised, among other posterior structures. In addition, also in the zebrafish embryo, Prep binds many loci in addition to ones associated with the anterior GRN and some of these additional sites are near genes that regulate other developmental processes known to involve TALE function. For instance, Prep peaks are found near genes involved in myogenesis (Figure 2D, E) and expression of myogenic genes is disrupted in TALE KD embryos, as the authors report in the Discussion section of the paper. Given all of these considerations, the statement above should be at least qualified and also toned down:

"These findings indicate that TALE factors control gene expression by regulating a chromatin transition – from repressive chromatin in blastula stage embryos to active chromatin in segmentation stage embryos – at a core set of genes encoding TFs that direct primarily anterior development in the zebrafish embryo."

4) Results, subsection “NF-Y proteins regulate TALE GRN expression and form complexes with TALE factors”, first paragraph: The authors describe a NF-Y motif that is specifically enriched at Prep3.5hpf peaks (Figure 7B). NF-Y is also maternally deposited in zebrafish (Figure 7—figure supplement 1A), consistent with a joint role for TALE and NF-Y factors at very early developmental stages. This is a critical finding, as it identifies a potential new cofactor for TALE proteins. The authors could better emphasize this exciting finding, for example drawing a parallel between the roles of NF-Y in zebrafish and in mouse embryonic development. In fact, also in the mouse NF-Y has been shown to have critical functions for very early embryonic development (Bhattacharya et al., 2003). This parallel would broaden the impact and breadth of the reported findings and would relate them also to other species, e.g. mammals.

5) Discussion: Given the wealth of high-quality results and novel findings that are reported in this interesting study, it would greatly help to add one additional figure – or figure panel – with a cartoon that summarizes and illustrates in a succinct manner the most salient findings. This would leave the reader with a strong and easy-to-remember 'take-home' message.

*Reviewer #3:*

The manuscript by Ladam et al. reports a ChIPseq analysis of Prep proteins at two developmental stages of the zebrafish embryo -blastula and segmentation- and the correlation of these data with transcriptome modifications associated to combined downregulation of Prep and Pbx factors using a morpholino approach. This work identifies a shift in binding preference by Prep factors between the two stages analyzed. While in the blastula the preferred binding site is a Prep-Pbx combined site, in the segmenting embryo the preference shifts to binding sites in which Prep binds to the Prep-alone binding site or the Pbx-Hox binding site (to which is can bind forming trimers). Classification of the blastula binding sites by their Histone Modification profile identifies a subset of targets that show repressive marks at the blastula stage and become active at the segmentation stage in a regionally-restricted manner. This class is enriched in genes whose expression and Histone modification profile is sensitive to the loss of Prep-Pbx function, indicating a pioneer function for Prep-Pbx in this tissue-specific genes.

In addition to this, authors analyze the occurrence binding sites for the pioneer transcription complex, which appear strongly associated to the Prep-Pbx binding motifs. They characterize association of Prep-Pbx to subunits of the NF-Y complex and they demonstrate that NF-Y activity is required for the transcriptional and epigenetic activation of Prep-Pbx responsive genes.

This is an important work mainly for two conclusions; one is that TALE transcription factors show stage-specific preference for binding sites defined by binding sequence. The data are compelling on this conclusion and suggest that it is the activation of tissue-specific transcriptional cofactors at later stages what directs this specificity.

The second finding is the identification of a cooperative complex between two pioneering complexes Prep-Pbx and NF-Y. This is an important finding, although some aspects remain only superficially explained or are not fully conclusive. My main concern on this point is that the authors claim to have discovered the function of the Prep-Pbx-NF-Y binding sites, however they do not perform any functional assay on this sequence. In my opinion, the data shown are only correlative with respect to this point. While elimination of Prep-Pbx function or elimination of NF-Y function affects a common set of targets, this does not demonstrate that the effect is through the action of these proteins as a complex on the Prep-Pbx-NF-Y binding sequence. Also, is the Prep/Pbx or NF-Y function different for those sites/genes in which only the Prep-Pbx sites are present versus those in which the Prep-Pbx-NF-Y site is found?. Also, given that the set of sensitive genes are tissue-specific and their activation associates with the colonization of nearby Prep-only and Pbx-Hox sites, could the function of these new sites be the one that is relevant for chromatin opening and transcriptional activation and not the interaction at the Prep-Pbx site?. Therefore, if possible, experiments in which the Prep-Pbx-NF-Y binding sequence is functionally analyzed for factor binding and enhancer activity should be included to make conclusions stronger.

In connection to this, a more comprehensive description of the prevalence of the Prep-Pbx and Prep-Pbx-NF-Y sites among the classes studied would help understanding whether there is specific association to subclasses. This would include the classification according to stage (blastula-specific, segmentation-specific and common) and MPAD classes. Also a detailed description of the occupancy of specific sites associated to the TALE GRN containing the Prep-Pbx-, Prep-Pbx-NFy, Prep-only or pbx-Hox binding sites and how these evolve between blastula and segmentation would be a very valuable addition to the manuscript.

---

## [Author Response]

Essential revisions:

*1) The authors must demonstrate that their model is correct by testing several of the identified elements by reporter assays in zebrafish embryos (or at the very least in cell lines* in vitro*), coupled with mutation of predicted TALE and NF-Y binding sites to address the importance of these sites for enhancer function.*

There are two types of elements identified by our analyses – one where Prep occupies monomeric HEXA binding sites associated with PBX:HOX motifs (this type of element is much more frequently occupied at 12hpf than at 3.5hpf) and one where Prep occupies DECA sites associated with NF-Y motifs (this is the predominantly occupied motif at 3.5hpf).

Elements containing HEXA+PBX:HOX motifs have been studied previously. Analyses of individual such elements in the mouse demonstrated that they act as enhancers and that the Prep, Pbx and Hox binding sites are essential for enhancer function (Pöpperl et al., 1995; Jacobs, Schnabel, and Cleary 1999; Ferretti et al., 2005; Ferretti et al., 2000; Di Rocco, Mavilio, and Zappavigna 1997; Manzanares et al., 2001; Tümpel et al., 2007; Wassef et al., 2008). Accordingly, we have previously shown that an element from the zebrafish *hoxb1a* locus (that contains a Prep+Pbx:Hox motif) drives gene expression in vivo and in vitro and that the TALE and Hox sites are essential (Choe et al., 2009; Vlachakis, Ellstrom, and Sagerström 2000). Furthermore, motif discovery in CNEs combined with functional testing in zebrafish recently identified HEXA+PBX:HOX motifs as being essential for enhancer activity (Grice et al., 2015; Parker et al., 2011). Hence, previous work has demonstrated that HEXA+PBX:HOX containing elements act as enhancers. This is now discussed in the Results subsection “Some TALE-occupied sites are associated with chromatin marks at blastula stages”, first paragraph. The revised manuscript now also includes additional data showing that the HEXA+PBX:HOX elements identified in our study are evolutionarily conserved (Figure 4—figure supplement 1A) and associated with known enhancer marks (Figure 4—figure supplement 1B). Finally, we now show that, of 74 enhancers known to be active in the zebrafish hindbrain (Grice et al., 2015; Parker et al., 2011), 19 (26%; Figure 4—figure supplement 1C) coincide with Prep_12hpf-only_ peaks. Since enhancer activity was assayed at 2-3 days of development, but our ChIP data is derived from 12hpf (and we applied a stringent FE>10 cutoff for the Prep peaks), this represents a substantial overlap. These results are now discussed in the aforementioned Results subsection. Hence, our analyses of HEXA+PBX:HOX elements are consistent with previous reports defining these elements as enhancers.

Since the DECA+NFY containing elements are novel, they have not been previously assayed for enhancer activity. Our data show that many such elements are associated with chromatin marks indicative of enhancers. In particular, our analysis in Figure 4A, B uses H3K4me1 (a modification that is well established as a mark of enhancers) to define MPADs at 4.5hpf and Figure 4F shows that many MPADs are marked by H3K27ac (a mark found at active enhancers) at 4.5hpf. Notably, many MPADs that lack H3K27ac at 4.5hpf, gain it over the next several hours (Figure 5A, B). In fact, even some Prep-bound sites that lack H3K4me1 at 4.5hpf (non-MPADs) gain enhancer marks during embryogenesis (Figure 5—figure supplement 1). Hence, many DECA+NFY elements appear to be (or become) associated with enhancer marks. We also find that these elements are enriched at regions of the genome that are conserved with mammals (revised Figure 4E), consistent with DECA+NFY elements playing an evolutionarily critical role. Importantly, even though DECA+NFY motifs are associated with enhancers, these motifs are not necessarily expected to be *sufficient* to convey enhancer activity. In fact, we have previously shown both in vivo and in vitro that TALE factors are not sufficient to drive gene expression (Choe, Ladam, and Sagerström 2014; Choe et al., 2009; Vlachakis, Choe, and Sagerström 2001), indicating that TALE factors alone do not mediate enhancer activity. Furthermore, Prep, Pbx and NF-Y are all ubiquitously expressed, but their binding is associated with genes that are expressed in a tissue-restricted manner, indicating that there must be additional regulatory input at enhancers for these genes. Accordingly, in the original manuscript, we did not propose that the DECA+NFY elements act as enhancers. Instead, our observation that TALE and NF-Y occupy many genomic sites prior to the appearance of enhancer marks (Figure 4A) and are required for the deposition of H3K27ac modifications at TALE-dependent genes (Figure 6F, 7F), led us to suggest that these factors are instead required at a step prior to establishment/action of enhancers (possibly in a ‘pioneer’ role). This is now clarified in the second paragraph of the Discussion and in a new model diagram in Figure 7H. Nevertheless, as requested by the reviewers, we have now cloned and tested genomic regions containing DECA+NFY sites in HEK293 cells. We have previously used transfection in HEK293 cells to demonstrate that HEXA+PBX:HOX containing elements from the zebrafish *hoxb1a* locus acts as an enhancer (Choe et al., 2009), so this is a reasonable system to assay zebrafish enhancers. We find that only one of the seven elements drives expression of a luciferase reporter (this is true also when TALE and NF-Y TFs are co-transfected with the reporter plasmids; data shown in Figure 7—figure supplement 1E), indicating that DECA+NFY elements may not function as classical enhancers. This is consistent with our proposed model that Prep, Pbx and NF-Y instead act at these elements to play an earlier role (possibly as pioneer factors) to permit the subsequent action of tissue-specific TFs (such as Hox TFs) at nearby enhancers. This model is clarified in the new Figure 7H.

2) The TALE knock-down phenotype needs to be better characterized with appropriate validation of the specificity of the morpholino.

[Additional details from reviewer 1: The TALE knock-down phenotype needs to be described/characterised in more detail, to provide a more comprehensive view of the developmental context and to validate the specificity of the morpholino cocktail. For instance, hindbrain segmentation/neuroanatomy and craniofacial morphology should be characterised in the triple morphants to provide more detailed evidence that the knockdown is as expected given the previously characterised MO phenotypes. I also suggest moving the justification for using MO's that is in the Figure 1—figure supplement 1 legend to the Materials and methods section or the main text, where it will be more prominent.]

Several published studies indicate that complete removal of Pbx or Prep function in zebrafish converges on the same phenotype (Pöpperl et al., 2000; Waskiewicz, Rikhof, and Moens 2002; Waskiewicz et al., 2001; Choe, Vlachakis, and Sagerström 2002; Deflorian et al., 2004). Accordingly, we expect that simultaneous disruption of Pbx and Prep function will also produce this phenotype. The published phenotype has been characterized morphologically, as well as by alcian blue staining for head cartilage, by immunostaining for hindbrain neurons and by in situ hybridization/microarray analysis for changes in gene expression. In the original submission of this manuscript, we show that our TALE KD phenotype is morphologically indistinguishable from the published phenotype (Figure 1—figure supplement 1A). We have now also assayed head cartilage formation and differentiation of hindbrain Mauthner neurons – we find that both are affected as expected based on the published reports (Figure 1—figure supplement 1B; this analysis also revealed that the pectoral fins are lost in our TALE KD embryos, as expected (Pöpperl et al., 2000)). This analysis is now covered in the Results subsection “TALE factors control a gene network regulating formation of anterior embryonic structures in zebrafish”. Lastly, we compared our RNA-seq data from TALE KD embryos to gene expression changes in two published reports (French et al., 2007; Deflorian et al., 2004). It is difficult to do a direct quantitative comparison since our RNA-seq was done at 12hpf after KD of both Prep and Pbx, while French et al. knocked down only Pbx and assayed gene expression (by microarray and in situ hybridization) at 18hpf and Deflorian et al. knocked down only Prep and assayed (only by in situ hybridization) at 20-24hpf. In spite of these caveats, of the 13 genes downregulated in French et al., seven were downregulated in our analysis and of the six genes downregulated in Deflorian et al., four were downregulated in our experiment (this is covered in the aforementioned subsection). We note that gene expression changes in TALE KD embryos become more pronounced over time (i.e., we see greater changes at 12hpf than at 6hpf), likely explaining why not all previously reported TALE-dependent genes are detected by our RNA-seq. In sum, we conclude that our TALE KD produces a phenotype very similar to the published phenotype resulting from Pbx or Prep knockdown.

As requested, the MO justification has been moved to the Materials and methods subsection “Interference with protein function in embryos”.

3) The authors need to explain the temporal discrepancies of all their assays.[Additional details from reviewer 2: The time-point analyzed are not fully consistent across these experimental approaches, e.g. RNA-Seq for controls and TALE KD zebrafish embryos is performed at 6hpf (early gastrula) and at 12hpf (segmentation stage); ChIP-seq for Prep1.1 is performed at 3.5hpf (blastula) and at 12hpf (segmentation stage); ATAC-Seq at 4hpf (blastula; available datasets); and ChIP-seq for chromatin marks at 4.5hpf (blastula; available datasets). While this could be somewhat concerning, the findings and the overall message emerging from the study are strong and do not appear to be weakened by the slight temporal discrepancies.]

Our analyses center on two timepoints – late blastula (3.5-4.5hpf) and early segmentation (12hpf). These timepoints were chosen because they represent a stage prior to zygotic gene expression (late blastula) and a stage coincident with the onset of tissue morphogenesis (early segmentation). These stages are now indicated in Figure 1A and the rationale for their selection is explained in the legend to Figure 1.

For the late blastula stage, we use chromatin ChIP-seq (4.5hpf) and ATAC-seq (4hpf) datasets for comparisons to our 3.5hpf Prep ChIP-seq data. The use of datasets from slightly different timepoints allowed us to use published datasets instead of replicating previous work. We agree with reviewer 2 that the difference between 3.5hpf, 4hpf and 4.5hpf is unlikely to affect our conclusions for two key reasons. 1) at these stages all zebrafish blastomere cells appear to be pluripotent and capable of differentiating into any tissue derivative (Ho and Kimmel 1993). 2) genome-wide analyses at these stages require large numbers of embryos (hundreds to thousands) that cannot be individually staged. Since zebrafish development is not perfectly synchronized, embryos will develop at slightly different speeds – even in a clutch from a single female. Therefore, when large numbers of embryos are collected, they are likely to contain a range of slightly different stages, meaning that ChIP-seq/ATAC-seq analyses done at 3.5hpf, 4hpf and 4.5hpf will actually have considerable overlap in the actual stages of the embryos analyzed. This fact is now mentioned in the Materials and methods subsection “Analysis of chromatin features”. We did not carry out RNA-seq at 3.5hpf because zygotic gene expression is not yet active at this timepoint. Instead, we selected the 6hpf timepoint (early gastrula), when zygotic gene expression is more robust. However, we do not detect any differentially expressed genes at 6hpf, so this RNA-seq dataset is not used to identify TALE-dependent genes.

4) The authors should provide clarifications/explanations regarding the statement that TALE GRN genes are significantly associated with Class 4 and Class 3, but not Class 1 or 2, MPADs (Figure 6A, B).[Additional details from Reviewer 2: The authors state that TALE GRN genes are significantly associated with Class 4 and Class 3, but not Class 1 or 2, MPADs (Figure 6A, B). However, RNA-seq (Figure 5C) shows that genes associated with Class 3 MPADs (and also Class 1 and 2 MPADs) are expressed at similar levels at 12hpf and 6hpf (Figure 5D). In contrast, Class 4 MPADs display higher levels of H3K27ac at 9hpf than at 4.5hpf (Figure 5A, B) and their associated genes show the greatest increase in expression between 6hpf and 12hpf. In addition, only class 4 MPADs showed a strong switch to an active chromatin state from 4.5hpf to 9hpf during zebrafish embryogenesis (Figure 5A, B), while class 3 MPADs did not exhibit any significant switch (Figure 5A, B). Collectively, these results are somewhat difficult to reconcile. The authors should qualify these findings and try to explain these differences. Are other factors necessary for activation of Class 3 MPADs? Or do the acetylation changes appear at a later time-point for Class 3 MPADs? Can other scenarios be envisaged? It would be helpful to add these considerations at least to the Discussion section of the paper.]

Our data show that TALE-dependent genes are associated with Class 3 and 4, but not Class 1 or 2 MPADs (Figure 6A). The reviewer comments that Class 3 MPAD-associated genes appear to behave more like Class 1 and 2 MPAD-associated genes in some regards and asks us to address this issue. However, our data indicate that, while the Class 3 MPAD-associated genes undergo smaller changes than Class 4-associated genes, they nevertheless show a greater increase in expression and higher H3K27ac levels than genes associated with Class 1 and 2 MPADs (Figure 5A-D), suggesting that they differ from Class 1 and 2 genes in these regards. These differences are now emphasized on lines 296-297. We do not know why Class 3 MPAD-associated genes show less pronounced signs of activation than Class 4 MPAD-associated genes, but we note that Class 3 MPAD-associated genes are more highly enriched for functions related to muscle differentiation (Figure 6C). This process is just beginning at the timepoint we used to assay gene expression (RNA-seq in Figure 5C, D) and H3K27ac (ChIP-seq in Figure 5A), suggesting that activation of Class 3 MPAD-associated genes may still be ongoing at this stage. This possibility is now discussed in the fifth paragraph of the Discussion.

5) The authors should include a final diagram depicting the model of Prep1.1/TALE DNA-binding dynamics across developmental time and how this relates to the activation of the components of the TALE-GRN, changes in chromatin state and interactions with co-factors.

A diagram is now included as Figure 7H.

6) Absolute statements should be avoided (e.g. "TALE factors control gene expression by regulating a chromatin transition… at a core set of genes encoding TFs that direct anterior development.")[Additional details from Reviewer 2: This statement is very strong and absolute, whereas it does not hold across the entire animal kingdom. In fact, in the mouse TALE factors substantially affect "posterior" development, as shown by the presence of severe posterior developmental defects in various mouse models with LOF for different TALE TFs. For example, in Pbx LOF mouse embryos it has been reported that the posterior axial skeleton, the hindlimb, and the urogenital system are severely compromised, among other posterior structures. In addition, also in the zebrafish embryo, Prep binds many loci in addition to ones associated with the anterior GRN and some of these additional sites are near genes that regulate other developmental processes known to involve TALE function. For instance, Prep peaks are found near genes involved in myogenesis (Figure 2D, E) and expression of myogenic genes is disrupted in TALE KD embryos, as the Authors report in the Discussion section of the paper. Given all of these considerations, the statement above should be at least qualified and also toned down: "These findings indicate that TALE factors control gene expression by regulating a chromatin transition – from repressive chromatin in blastula stage embryos to active chromatin in segmentation stage embryos – at a core set of genes encoding TFs that direct primarily anterior development in the zebrafish embryo."]

The statement referenced by reviewer 2 has been revised as requested (subsection “TALE factors control the chromatin state at Class 4 MPADs associated with the anterior GRN”). We have also reviewed the text to make sure that our statements do not imply general conclusions, unless warranted.

Reviewer #1:

[…] i) The TALE knock-down phenotype needs to be described/characterised in more detail, to provide a more comprehensive view of the developmental context and to validate the specificity of the morpholino cocktail. For instance, hindbrain segmentation/neuroanatomy and craniofacial morphology should be characterised in the triple morphants to provide more detailed evidence that the knockdown is as expected given the previously characterised MO phenotypes. I also suggest moving the justification for using MO's that is in the Figure 1—figure supplement 1 legend to the Materials and methods section or the main text, where it will be more prominent.

This issue is addressed under Essential revisions – point 2 above.

*ii) An assumption made is that Prep1.1-bound sites, or at least a sub-set of them, represent enhancer elements. The authors must demonstrate that this is true by testing a few such elements by reporter assay in zebrafish embryos, coupled with mutation of predicted TALE and NF-Y sites to address the importance of these sites for enhancer function. This can be done relatively quickly by transient transgenesis, is frequently used for mechanistic dissection of cis-regulatory elements, and will provide crucial evidence for the functionality of these putative enhancers.*

This issue is addressed under Essential revisions – point 1 above.

iii) The authors use data from mouse ESCs to infer evolutionary conservation of the TALE GRN and of TALE-NF-Y co-localised binding, which expands the scope beyond zebrafish. A complementary approach is to address how many Prep1.1-bound peaks overlap with fish-mammal conserved non-coding elements that have been described in the literature. It is also worth checking if any are homologous to elements in the VISTA enhancer browser and have been experimentally validated in transgenic mice. This is straightforward to do and could potentially add weight to the argument that these interactions are evolutionarily conserved.

This issue was addressed as part of Essential revisions – point 1 above. Specifically, the revised Figure 4E now shows that MPAD sites (particularly the Class 4 sites that are found near TALE-dependent developmental regulators) coincide with genomic regions conserved among vertebrates. In addition, we selected a subset of elements for more in depth analyses of conservation and this data is now presented in Figure 4—figure supplement 1D. Lastly, we find that many enhancers shown to be active in the hindbrain (Grice et al., 2015; Parker et al., 2011), coincide with Preppeaks (Figure 4—figure supplement 1C and subsection “Some TALE-occupied sites are associated with chromatin marks at blastula stages”, first paragraph).

iv) This manuscript would really benefit from a final diagram depicting the model of Prep1.1/TALE DNA-binding dynamics across developmental time and how this relates to the activation of the components of the TALE-GRN, changes in chromatin state and interactions with co-factors.

A diagram is now included as Figure 7H.

Reviewer #2:

[…] Concerns that should be addressed:1) Results – General Consideration:[…] However, it might be useful to underscore throughout the text and in the Discussion that TALE factors adopt distinct mechanistic strategies in zebrafish blastula and early gastrula versus segmentation stages; in other words to simply cluster together the functions of TALE TFs in blastula and early gastrula within one single group (comprising 3.5, 4, 4.5, and 6 hpf). To this end, it would help to slightly modify the cartoon illustrating the subsequent zebrafish developmental stages in Figure 1A. The authors could group [blastula stages and early gastrula stages] within one single bracket or inside one single box andthe [segmentation stages] inside another bracket or box. Accordingly, this clustering could be clarified in the figure legend.

This issue is addressed under Essential revisions – point 3 above.

2) Results, subsection “TALE factors control the chromatin state at Class 4 MPADs associated with the anterior GRN”: The authors state that TALE GRN genes are significantly associated with Class 4 and Class 3, but not Class 1 or 2, MPADs (Figure 6A, B). However, RNA-seq (Figure 5C) shows that genes associated with Class 3 MPADs (and also Class 1 and 2 MPADs) are expressed at similar levels at 12hpf and 6hpf (Figure 5D). In contrast, Class 4 MPADs display higher levels of H3K27ac at 9hpf than at 4.5hpf (Figure 5A, B) and their associated genes show the greatest increase in expression between 6hpf and 12hpf. In addition, only class 4 MPADs showed a strong switch to an active chromatin state from 4.5hpf to 9hpf during zebrafish embryogenesis (Figure 5A, B), while class 3 MPADs did not exhibit any significant switch (Figure 5A, B). Collectively, these results are somewhat difficult to reconcile. The authors should qualify these findings and try to explain these differences. Are other factors necessary for activation of Class 3 MPADs? Or do the acetylation changes appear at a later time-point for Class 3 MPADs? Can other scenarios be envisaged? It would be helpful to add these considerations at least to the Discussion section of the paper.

This issue is addressed under Essential revisions – point 4 above.

3) Results: The authors state: "These findings indicate that TALE factors control gene expression by regulating a chromatin transition – from repressive chromatin in blastula stage embryos to active chromatin in segmentation stage embryos – at a core set of genes encoding TFs that direct anterior development."This statement is very strong and absolute, whereas it does not hold across the entire animal kingdom. In fact, in the mouse TALE factors substantially affect "posterior" development, as shown by the presence of severe posterior developmental defects in various mouse models with LOF for different TALE TFs. For example, in Pbx LOF mouse embryos it has been reported that the posterior axial skeleton, the hindlimb, and the urogenital system are severely compromised, among other posterior structures. In addition, also in the zebrafish embryo, Prep binds many loci in addition to ones associated with the anterior GRN and some of these additional sites are near genes that regulate other developmental processes known to involve TALE function. For instance, Prep peaks are found near genes involved in myogenesis (Figure 2D, E) and expression of myogenic genes is disrupted in TALE KD embryos, as the authors report in the Discussion section of the paper. Given all of these considerations, the statement above should be at least qualified and also toned down:"These findings indicate that TALE factors control gene expression by regulating a chromatin transition – from repressive chromatin in blastula stage embryos to active chromatin in segmentation stage embryos – at a core set of genes encoding TFs that direct primarily anterior development in the zebrafish embryo."

This issue is addressed under Essential revisions – point 6 above.

4) Results, subsection “NF-Y proteins regulate TALE GRN expression and form complexes with TALE factors”, first paragraph: The authors describe a NF-Y motif that is specifically enriched at Prep3.5hpf peaks (Figure 7B). NF-Y is also maternally deposited in zebrafish (Figure 7—figure supplement 1A), consistent with a joint role for TALE and NF-Y factors at very early developmental stages. This is a critical finding, as it identifies a potential new cofactor for TALE proteins. The authors could better emphasize this exciting finding, for example drawing a parallel between the roles of NF-Y in zebrafish and in mouse embryonic development. In fact, also in the mouse NF-Y has been shown to have critical functions for very early embryonic development (Bhattacharya et al., 2003). This parallel would broaden the impact and breadth of the reported findings and would relate them also to other species, e.g. mammals.

A more extensive discussion of previous work on NF-Y in embryogenesis (including the Bhattacharya reference) is now included in the Results subsection “NF-Y proteins regulate TALE GRN expression and form complexes with TALE factors”, second paragraph.

5) Discussion: Given the wealth of high-quality results and novel findings that are reported in this interesting study, it would greatly help to add one additional figure – or figure panel – with a cartoon that summarizes and illustrates in a succinct manner the most salient findings. This would leave the reader with a strong and easy-to-remember 'take-home' message.

A diagram is now included as Figure 7H.

Reviewer #3:

[…] *This is an important work mainly for two conclusions; one is that TALE transcription factors show stage-specific preference for binding sites defined by binding sequence. The data are compelling on this conclusion and suggest that it is the activation of tissue-specific transcriptional cofactors at later stages what directs this specificity.*

The second finding is the identification of a cooperative complex between two pioneering complexes Prep-Pbx and NF-Y. This is an important finding, although some aspects remain only superficially explained or are not fully conclusive. My main concern on this point is that the authors claim to have discovered the function of the Prep-Pbx-NF-Y binding sites, however they do not perform any functional assay on this sequence. In my opinion, the data shown are only correlative with respect to this point. While elimination of Prep-Pbx function or elimination of NF-Y function affects a common set of targets, this does not demonstrate that the effect is through the action of these proteins as a complex on the Prep-Pbx-NF-Y binding sequence.

We have now carried out a functional analysis of these binding sites (see Essential revisions – point 1 above).

In the original manuscript, we were careful not to claim that a Prep:Pbx:NFY complex represents the functional unit controlling the TALE GRN – since this is difficult to prove experimentally. Our data show 1) that TALE and NF-Y TFs occupy binding sites (DECA and CCAAT) that are close to each other (average distance = 20bp), 2) that these TFs regulate a shared set of genes and 3) that they can form a complex. Hence, “We conclude that NF-Y binds adjacent to TALE factors at DECA sites and that both factors are required for regulation of the TALE GRN, possibly by functioning in a complex.”

Also, is the Prep/Pbx or NF-Y function different for those sites/genes in which only the Prep-Pbx sites are present versus those in which the Prep-Pbx-NF-Y site is found?

We partially addressed this point in the original manuscript (new Figure 7—figure supplement 1C), showing that TALE-occupied sites with adjacent NF-Y motifs have higher levels of H3K27ac, reduced nucleosome occupancy and reduced levels of H3K27me3 compared to ones lacking NF-Y motifs. However, these effects are relatively subtle and we did not examine them further. The main limitation with this avenue of inquiry is that we do not have NF-Y ChIP-seq data in zebrafish. Hence, we do not know which NF-Y sites are occupied (our ChIP-qPCR analysis in Figure 7C indicates that ~60% of NF-Y sites are occupied), which makes it difficult to draw conclusions about NF-Y activity at these sites.

Also, given that the set of sensitive genes are tissue-specific and their activation associates with the colonization of nearby Prep-only and Pbx-Hox sites, could the function of these new sites be the one that is relevant for chromatin opening and transcriptional activation and not the interaction at the Prep-Pbx site?.

The experiments in Figure 6F and 7F, which demonstrate that TALE and NF-Y are required for increased H3K27 acetylation, specifically assay the chromatin state at MPADs (that contain DECA+NFY motifs) and were done at a stage (9hpf) prior to expression of most tissue-specific genes. This suggests that TALE and NF-Y acting at DECA+NFY motifs play an important role.

Therefore, if possible, experiments in which the Prep-Pbx-NF-Y binding sequence is functionally analyzed for factor binding and enhancer activity should be included to make conclusions stronger.

This issue is addressed under Essential revisions – point 1 above.

In connection to this, a more comprehensive description of the prevalence of the Prep-Pbx and Prep-Pbx-NF-Y sites among the classes studied would help understanding whether there is specific association to subclasses. This would include the classification according to stage (blastula-specific, segmentation-specific and common) and MPAD classes. Also a detailed description of the occupancy of specific sites associated to the TALE GRN containing the Prep-Pbx-, Prep-Pbx-NFy, Prep-only or pbx-Hox binding sites and how these evolve between blastula and segmentation would be a very valuable addition to the manuscript.

We do not find any significant differences in the distribution of DECA versus DECA+NFY sites among the various regulatory elements (MPADs and non-MPADs), nor do we find a difference in their association with TALE GRN genes. This is now shown in Figure 7—figure supplement 1D and discussed in the second paragraph of the subsection “NF-Y proteins regulate TALE GRN expression and form complexes with TALE factors”.